# Risks of specific congenital anomalies in offspring of women with diabetes: A systematic review and meta-analysis of population-based studies including over 80 million births

**Tie-Ning Zhang**[1,2,3]**, Xin-Mei Huang**[4]**, Xin-Yi Zhao**[1,2,3]**, Wei Wang**[3]**, Ri Wen**[3]**, Shan-Yan Gao** [1,2] *

**1** Department of Clinical Epidemiology, Shengjing Hospital of China Medical University, Shenyang, China, **2** Clinical Research Center, Shengjing Hospital of China Medical University, Shenyang, China, **3** Department of Pediatrics, Shengjing Hospital of China Medical University, Shenyang, China, **4** Department of Endocrinology, Shanghai Fifth People's Hospital, Fudan University, Shanghai, China

* gaosy@sj-hospital.org

**Data Availability Statement:** The metadata underlying the reported analyses have been deposited in Zenodo (DOI: 10.5281/zenodo.

## Abstract

### Background

Pre-gestational diabetes mellitus (PGDM) has been known to be a risk factor for congenital heart defects (CHDs) for decades. However, the associations between maternal PGDM and gestational diabetes mellitus (GDM) and the risk of specific types of CHDs and congenital anomalies (CAs) in other systems remain under debate. We aimed to investigate type-specific CAs in offspring of women with diabetes and to examine the extent to which types of maternal diabetes are associated with increased risk of CAs in offspring.

### Methods and findings

We searched PubMed and Embase from database inception to 15 October 2021 for population-based studies reporting on type-specific CAs in offspring born to women with PGDM (combined type 1 and 2) or GDM, with no limitation on language. Reviewers extracted data for relevant outcomes and performed random effects meta-analyses, subgroup analyses, and multivariable meta-regression. Risk of bias appraisal was performed using the Cochrane Risk of Bias Tool. This study was registered in PROSPERO (CRD42021229217). Primary outcomes were overall CAs and CHDs. Secondary outcomes were type-specific CAs. Overall, 59 population-based studies published from 1990 to 2021 with 80,437,056 participants met the inclusion criteria. Of the participants, 2,407,862 (3.0%) women had PGDM and 2,353,205 (2.9%) women had GDM. The meta-analyses showed increased risks of overall CAs/CHDs in offspring born to women with PGDM (for overall CAs, relative risk [RR] = 1.99, 95% CI 1.82 to 2.17, $P < 0.001$; for CHDs, RR = 3.46, 95% CI 2.77 to 4.32, $P < 0.001$) or GDM (for overall CAs, RR = 1.18, 95% CI 1.13 to 1.23, $P < 0.001$; for CHDs, RR = 1.50, 95% CI 1.38 to 1.64, $P < 0.001$). The results of the meta-regression analyses

5783967). https://doi.org/10.5281/zenodo.
5783967.

**Funding:** The authors received no specific funding
for this work.

**Competing interests:** The authors have declared
that no competing interests exist.

**Abbreviations:** AMPK, AMP-activated protein
kinase; CHD, congenital heart defect; CA,
congenital anomaly; CAKUT, congenital
abnormalities of the kidney and the urinary tract;
CI, confidence interval; GDM, gestational diabetes
mellitus; MOOSE, Meta-analysis Of Observational
Studies in Epidemiology; OS, oxidative stress;
PGDM, pre-gestational diabetes mellitus; PRISMA,
Preferred Reporting Items for Systematic Reviews
and Meta-Analyses; RR, relative risk; ROBINS-I,
Risk of Bias in Non-randomized Studies–of
Interventions; ROS, reactive oxygen species.

showed significant differences in RRs of CAs/CHDs in PGDM versus GDM (all $P$ < 0.001).
Of the 23 CA categories, excluding CHD-related categories, in offspring, maternal PGDM
was associated with a significantly increased risk of CAs in 21 categories; the corresponding
RRs ranged from 1.57 (for hypospadias, 95% CI 1.22 to 2.02) to 18.18 (for holoprosence-
phaly, 95% CI 4.03 to 82.06). Maternal GDM was associated with a small but significant
increase in the risk of CAs in 9 categories; the corresponding RRs ranged from 1.14 (for
limb reduction, 95% CI 1.06 to 1.23) to 5.70 (for heterotaxia, 95% CI 1.09 to 29.92). The
main limitation of our analysis is that some high significant heterogeneity still persisted in
both subgroup and sensitivity analyses.

## Conclusions

In this study, we observed an increased rate of CAs in offspring of women with diabetes and
noted the differences for PGDM versus GDM. The RRs of overall CAs and CHDs in off-
spring of women with PGDM were higher than those in offspring of women with GDM.
Screening for diabetes in pregnant women may enable better glycemic control, and may
enable identification of offspring at risk for CAs.

## Author summary

### Why was this study done?

- It is controversial whether maternal pre-gestational or gestational diabetes affects spe-
  cific types of congenital heart defects (CHDs) and congenital anomalies (CAs) in other
  systems.

- Comprehensive estimates of the risks of specific CAs for offspring of women with
  maternal diabetes are needed to counsel patients and for public health purposes.

### What did the researchers do and find?

- To the best of our knowledge, this is the first comprehensive systematic review and
  meta-analysis of population-based studies of over 80 million participants that demon-
  strates an increased risk of type-specific CAs, especially CHDs, in offspring born to
  women with pre-gestational or gestational diabetes.

- Our study shows that maternal pre-gestational diabetes is associated with a significant
  increase in the risk of 38 out of 45 categories of CAs in offspring, while maternal gesta-
  tional diabetes is associated with a small but significant increase in the risk of 16 out of
  the 45 categories.

- The corresponding relative risks (RRs) of overall CAs/CHDs in offspring of women
  with pre-gestational diabetes are higher than those in offspring of women with gesta-
  tional diabetes, with no overlap in the 95% CIs.

## What do these findings mean?

- In this study, we observed that there is an increased rate of CAs in offspring of women with maternal diabetes and noted the differences between pre-gestational and gestational diabetes.

- Considering the substantial rise in the prevalence of maternal diabetes over recent decades, the expectation that this prevalence will continue to increase, the number of pregnancies worldwide, and the significant individual and global burdens associated with CAs, it is crucial that healthcare providers are aware of this association and can identify women and offspring who are at risk.

## Introduction

Currently, the global prevalence of diabetes is increasing among women of reproductive age [1,2]. A diabetic intrauterine environment can cause placental dysfunction and hormone alterations, which could lead to various congenital anomalies (CAs) in offspring of women with diabetes [1]. Notably, pre-gestational diabetes mellitus (PGDM, which includes type 1 and 2 diabetes) has been known to be a risk factor for congenital heart defects (CHDs) for decades [3]. However, there is controversy among current research regarding the association between maternal PGDM and the risk of specific types of CHDs and other CAs of the nervous, digestive, genitourinary, and musculoskeletal systems [4–7]. Further studies are thus needed for clarification of this risk.

Gestational diabetes mellitus (GDM), which is defined as any degree of glucose intolerance with onset or first recognition during pregnancy, is one of the most common complications of pregnancy and affects up to 9%–26% of the obstetric population [8,9]. Similar to PGDM, GDM also has a considerable impact on the health outcomes of the mother and infant during pregnancy, delivery, and beyond. Recently, an increasing number of studies have concentrated on evaluating the risks of specific types of CAs in offspring born to women with GDM [4,5,10–12]. The early period of organogenesis, which occurs during the third to eighth week of gestation, is an important time for organ development. However, hyperglycemia associated with GDM occurs after this critical early window for organogenesis. Therefore, the question as to whether there is an association between GDM and the risk of specific types of CAs in offspring remains.

Previous meta-analyses have mainly focused on the associations between maternal diabetes and CHDs in offspring, and little is known about the influence of maternal diabetes on other specific types of CAs [13,14]. Additionally, new data from population-based studies of more than 36 million births have provided solid estimates of the risk of CHDs in offspring of women with diabetes [4,10–12]. This considerable amount of data could also be used to explore the association between maternal diabetes and other types of CAs. Currently, a quantitative summary of population-based studies on the associations between maternal diabetes (pre-gestational or gestational) and type-specific CAs in offspring is lacking. Comprehensive estimates of the risk of specific CAs associated with maternal diabetes are needed to counsel patients and for public health purposes. Moreover, it is essential that estimates are provided according to different types of maternal diabetes, given the diversity in etiology, treatment, and prognosis.

We performed a detailed systematic review and large-scale meta-analysis to summarize and quantify the existing population-based data on type-specific CAs in offspring of women with diabetes. Furthermore, we examined the extent to which specific types of maternal diabetes (i.e., pre-gestational or gestational) are associated with increased risk of CAs in offspring.

## Methods

We performed a literature search in accordance with the Preferred Reporting Items for Systematic Reviews and Meta-Analyses (PRISMA) [15] and Meta-analysis Of Observational Studies in Epidemiology (MOOSE) guidelines [16] (see S1 Text). Before study selection, the protocol for this review was registered in PROSPERO, registration number CRD42021229217 (S1 Protocol).

### Search strategy and inclusion criteria

We searched PubMed and Embase from database inception to 15 October 2021. The search strategy combined Medical Subject Headings (MeSH) and Embase subject heading (Emtree) terms with other unindexed or free text terms, with no limitation on language. Details of the full search strategy are provided in S1 Text. Reference lists of retrieved articles and previous systematic and narrative reviews were searched manually to retrieve all relevant documents. Duplicate citations were removed.

Population-based cross-sectional, case–control, and cohort studies that reported original data were eligible for inclusion if they (1) reported any CAs in offspring born to women with diabetes (i.e., pre-gestational [combined type 1 and 2] or gestational diabetes), (2) had a comparison group that included mothers without diabetes, and (3) provided sufficient data from which a risk estimate could be calculated if a risk estimate was not reported. All conference abstracts, guidelines, case reports, case series, commentaries, letters, and animal studies were excluded.

Two independent authors (S-YG and T-NZ) reviewed the titles and abstracts to identify any relevant studies. The full texts of potentially eligible studies that appeared to meet the inclusion criteria were then obtained and independently evaluated by the 2 reviewers. Any disagreement was settled by consensus among all authors. If multiple studies were derived from the same dataset and reported the same associated outcome, the study with the most complete findings or the greatest number of participants was included for analysis. The literature review and study selection process referenced the PRISMA flowchart (Fig 1). When information needed for inclusion in the analyses was missing, the Library of Shengjing Hospital of China Medical University helped us get full access of the relevant data.

### Data extraction

A standardized, pre-designed spreadsheet was used for extracting data from the included studies. Study quality and synthesis of evidence were assessed. The following data were recorded in the spreadsheet: first author, publication year, geographic location, study period, study design, data source, type of diabetes, sample size, types of birth, ascertainment of exposure, definition of outcome, outcome risk estimates and 95% confidence intervals (CIs), and adjusted confounders.

Primary outcomes were the rates of overall CAs and type-specific CHDs (heterotaxia, conotruncal defects, atrioventricular septal defect, anomalous pulmonary venous return, left ventricular outflow tract defect, right ventricular outflow tract defect, septal defects, and single ventricle). Secondary outcomes were the rates of other type-specific CAs (involving the nervous system; eye, ear, face, and neck; orofacial clefts; digestive system; abdominal wall; genitourinary system; and musculoskeletal system). S1 Table shows the definitions of these outcomes.

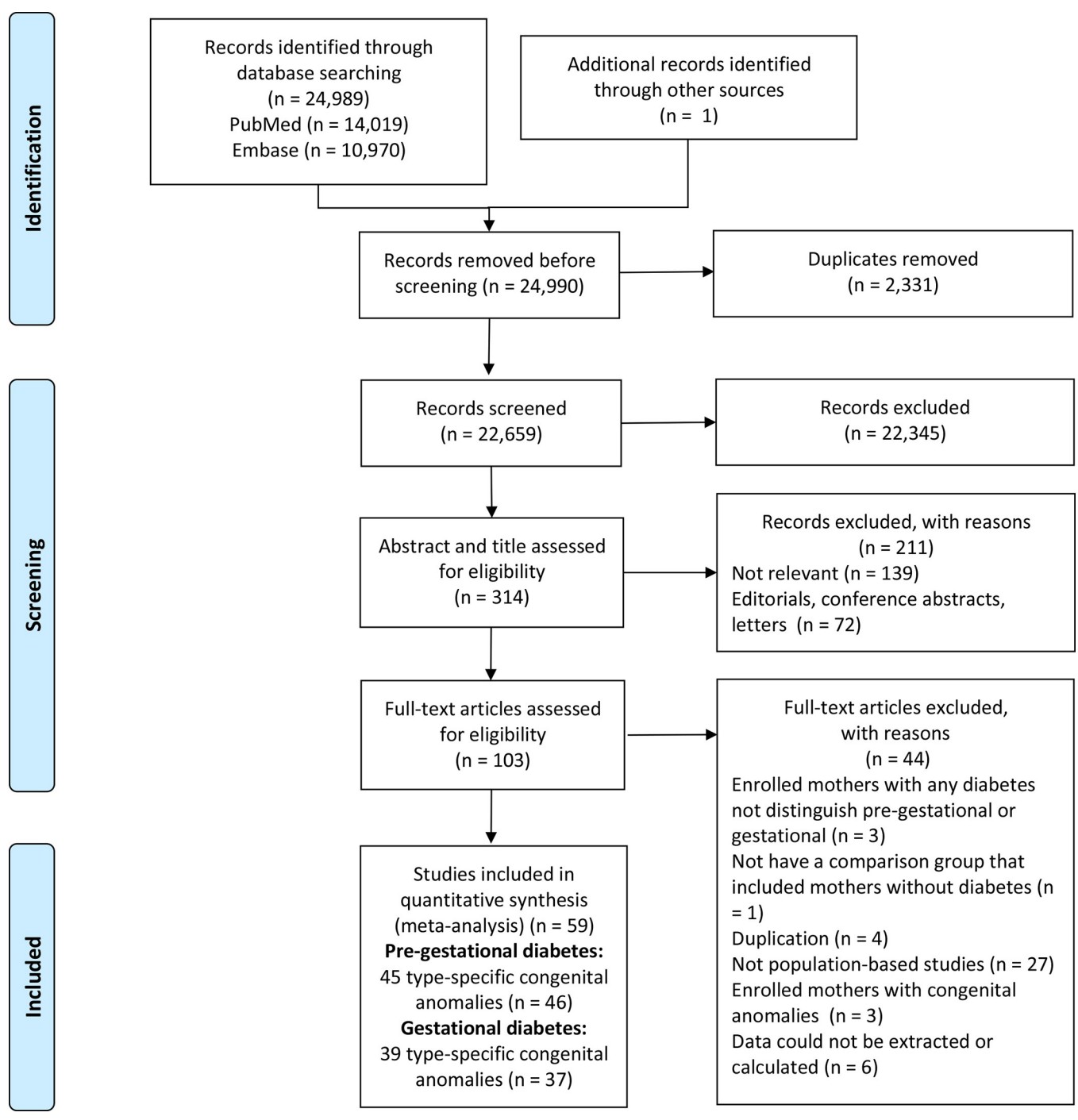

**Fig 1. Flowchart of selection of studies included in the meta-analysis.**

Two authors (T-NZ and S-YG) independently performed data extraction according to the Cochrane Handbook guidelines [17]. Findings were reported according to PRISMA [15] and MOOSE guidance [16]. Any disagreement was settled by consensus among all authors. For studies that did not report any adjusted effect sizes, the crude risk estimate was used. If an included study reported several risk estimates, we extracted the fully adjusted effect sizes. For

studies that reported the risk estimates of CAs stratified by isolated and multiple statuses, we used the effect sizes of the isolated CAs. Because odds ratios, prevalence rate ratios, and hazard ratios are excellent approximations of risk ratios in the case of rare outcomes [18], all risk estimates are referred to and reported as relative risks (RRs) for simplicity. If an included study lacked required data, we asked for help from the Library of Shengjing Hospital of China Medical University to get the missing information.

## Risk of bias and study quality

The risk of bias assessment was conducted by T-NZ and S-YG using the Risk of Bias in Non-randomized Studies–of Interventions (ROBINS-I) tool [19]. This tool comprises 7 domains: bias due to confounding, bias in the selection of participants, bias in the classification of interventions, bias due to deviations from intended interventions, bias due to missing data, bias in measurement of outcomes, and bias in the selection of the reported result. We rated the possible risk of bias in each of the 7 domains as low risk, moderate risk, serious risk, critical risk, or no information for each available outcome of each included study.

## Statistical analysis

For studies that reported effect sizes separately, the results were pooled using a fixed effects model to obtain an overall estimate and then included in the pooled effect size in the meta-analysis. The effective count method proposed by Hamling et al. [20] was used to recalculate the effect sizes. If a selected study did not include an effect size, the unadjusted risk estimate and 95% CI were calculated from the raw data for simplicity using EpiCalc 2000 (https://en.freedownloadmanager.org/Windows-PC/EpiCalc-2000-FREE.html). Estimates were pooled using the DerSimonian and Laird random effects model to calculate summarized RRs and 95% CIs [21], in which $I^2$ values were calculated as indicators of heterogeneity. $I^2$ values of $\leq$25%, 26%–50%, 51%–74%, and $\geq$75% were considered to indicate no, low, moderate, and high heterogeneity between the included studies, respectively [22]. For the primary outcomes of the study, subgroup analyses were undertaken to explore causes of heterogeneity: by region (Europe, North/South America, or Asia-Pacific), year of enrollment (categorized using the median as the cutoff value: before 1997 or in or after 1997), number of participants (categorized using the median as the cutoff value: <282,260 or $\geq$282,260), and adjustment for confounders (i.e., maternal age, race/ethnicity, body mass index, education, smoking/alcohol consumption, parity, and pregnancy complications). Heterogeneity between subgroups was evaluated by meta-regression analysis if data were reported in more than 10 studies, following to the Cochrane guidelines [23]. Meta-regression analyses were also used to examine the extent to which the types of maternal diabetes (i.e., pregestational or gestational) are associated with increased risk of overall CAs/CHDs in offspring. Publication bias was examined by inspecting funnel plots for the outcomes and was further evaluated with Begg's test [24] and Egger's test [25] if sufficient studies existed ($n \geq 10$) [17]. A sensitivity analysis was undertaken to explore the association of each study with the overall pooled estimate. Statistical analyses were conducted using Stata version 13.0 (StataCorp, College Station, Texas). A 2-tailed $P$ value less than 0.05 was considered statistically significant.

## Results

### Search results and study characteristics

We identified 24,989 potentially eligible articles in PubMed and Embase through the search strategy plus 1 additional article through hand searching. Of these, 2,331 records were duplicates (Fig 1). In total, 103 articles qualified for full-text review based on title and abstract

screening. Of these, an additional 44 articles were excluded for the following reasons: 3 studies enrolled mothers with any type of diabetes and did not distinguish pre-gestational and gestational diabetes, 1 study did not have a comparison group that included mothers without diabetes, 4 studies were derived from the same dataset and reported the same associated outcome as another included study, 3 studies enrolled mothers with CAs, 6 studies included data that could not be extracted or calculated, and 27 studies were not population-based (S2 Table). Finally, 59 population-based studies (published from 1990 to 2021) that met all eligibility criteria contributed to the quantitative synthesis and included a total of 80,437,056 participants (range of participants per study: 155 to 29,211,974) for analysis. Of these, 2,407,862 (3.0%) women had PGDM and 2,353,205 (2.9%) women had GDM; 879,156 cases of overall CAs and 350,051 cases of CHDs were observed. S3 Table gives the method of ascertainment of maternal diabetes of the included studies. Of the 59 studies included, there were 27 studies from the European region (United Kingdom [26–30], Finland [31,32], Sweden [10,33–37], Denmark [10,28,35,38,39], Norway [6,28,40,41], Hungary [7,42–45], Germany [28,46], Netherlands [28], Belgium [28], Wales [28,30], Ireland [28,30], Switzerland [28], France [28,47], Italy [28,48,49], Spain [28], Portugal [28], and Malta [28]), 25 studies from North/South America (United States [4,5,11,12,14,50–57], Canada [50,58–68], and Brazil [69]), and 7 studies from the Asian-Pacific region (China [70–72], Russia [73], Australia [74], and Qatar [75,76]). Table 1 summarizes the characteristics of the included studies, and a more detailed breakdown can be found in S4 Table.

## Bias assessment

We assessed the risk of bias for 34 of 59 included studies using ROBINS-I. The assessments are summarized for primary outcomes in Figs A–D in S1 Fig. None of the included studies were

**Table 1. Summary characteristics of included studies.**

| Characteristic | Number of studies (number of participants) |
|---|---|
| **Eligible studies** | 59 (80,437,056) |
| **Region** | |
| Europe | 27 (19,297,559) |
| North/South America | 25 (52,645,605) |
| Asia-Pacific | 7 (8,493,892) |
| **Year of enrollment** | |
| Before 1997 | 27 (14,896,852) |
| In or after 1997 | 32 (65,540,204) |
| **Type of maternal diabetes** | |
| Pre-gestational diabetes | 46* (2,407,862) |
| Type 1 diabetes | 11 (285,859) |
| Type 2 diabetes | 7 (294,525) |
| Gestational diabetes | 37 (2,353,205) |
| **Primary outcomes** | |
| Overall CAs | 24 (879,156) |
| CHDs | 23 (350,051) |

The median (range) number of participants per study was 282,260 (155 to 29,211,974). Studies included 45 type-specific CAs. CA, congenital anomaly; CHD, congenital heart defect.

*Of 46 studies, 28 studies also reported results on gestational diabetes

7 studies also reported results on type 1 and type 2 diabetes.

rated with a low risk of bias in all domains. The main causes of serious or critical bias risk according to ROBINS-I were weaknesses in the confounding bias domain, selection of participant bias domain, and missing data bias domain.

## Exposure to PGDM/GDM and overall CAs in offspring

We first explored whether there was an association between maternal diabetes and overall CAs (not including CHDs) in offspring. Nineteen studies investigated the relationship between maternal PGDM and overall CAs in offspring [4,11,12,27,29,38,44,46,49,56,58,61–63,65,67,70,74,76], and 15 studies investigated the relationship between maternal GDM and overall CAs in offspring [4,11,12,33,39,44,48,56,58,61,62,65,67,69,76]. Our results suggested that maternal PGDM was associated with overall CAs in offspring (RR = 1.99, 95% CI 1.82 to 2.17, $I^2$ = 90.0%, $P$ < 0.001; Fig 2), with no evidence of publication bias (Begg's $P$ = 0.88, Egger's $P$ = 0.30; Fig E in S1 Fig). A similar association was observed for overall CAs in offspring of women with type 1 diabetes

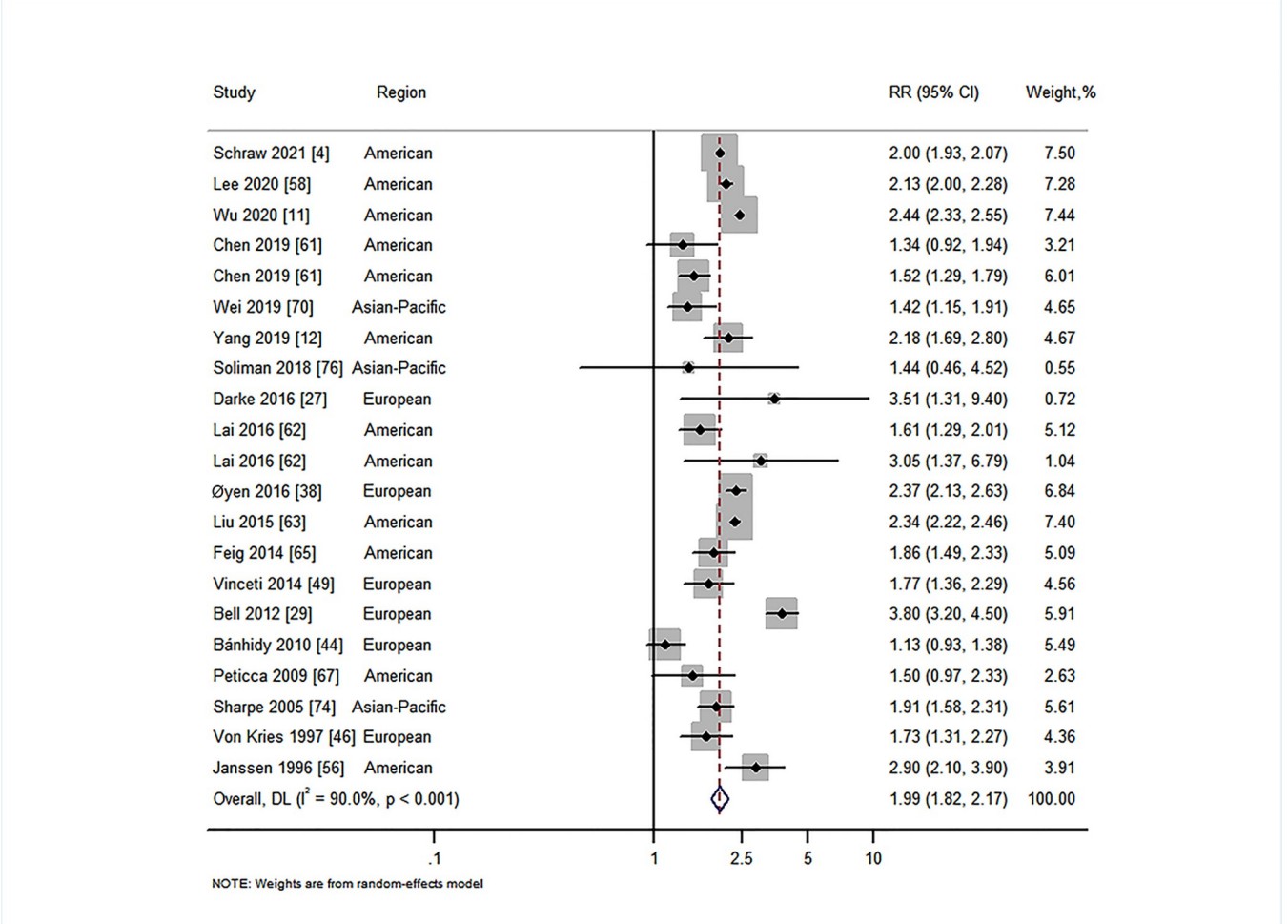

**Fig 2. Forest plot of the RRs in population-based studies for maternal pre-gestational diabetes and the risk of overall congenital anomalies (RR = 1.99, 95% CI 1.82 to 2.17, $I^2$ = 90.0%, $P$ < 0.001).** Analytical weights are from random effects meta-analysis. Grey boxes represent study estimates; their size is proportional to the respective analytical weight. Lines through the boxes represent the 95% CIs around the study estimates. The diamond represents the mean estimate and its 95% CI. The vertical red dashed line indicates the mean estimate. CI, confidence interval; DL, DerSimonian and Laird random effects model; RR, relative risk.

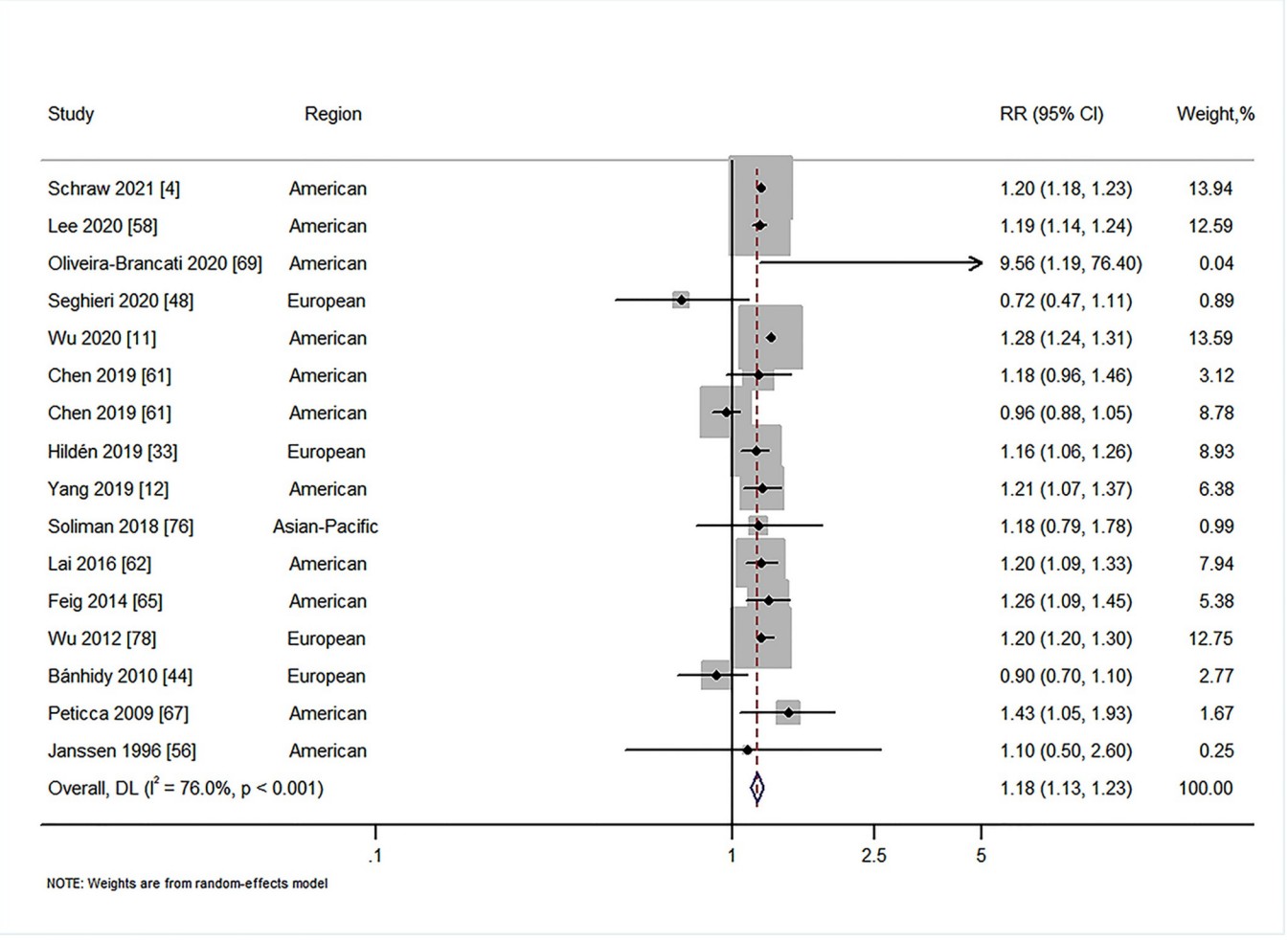

**Fig 3. Forest plot of the RRs in population-based studies for maternal gestational diabetes and the risk of overall congenital anomalies (RR = 1.18, 95% CI 1.13 to 1.23, $I^2$ = 76.0%, $P$ < 0.001).** Analytical weights are from random effects meta-analysis. Grey boxes represent study estimates; their size is proportional to the respective analytical weight. Lines through the boxes represent the 95% CIs around the study estimates. The diamond represents the mean estimate and its 95% CI. The vertical red dashed line indicates the mean estimate. CI, confidence interval; DL, DerSimonian and Laird random effects model; RR, relative risk.

(RR = 2.03, 95% CI 1.66 to 2.48, $I^2$ = 82.5%, $P$ < 0.001; Fig K1 in S1 Fig) and in offspring of women with GDM (RR = 1.18, 95% CI 1.13 to 1.23, $I^2$ = 76.0%, $P$ < 0.001; Fig 3), with no evidence of publication bias (Begg's $P$ = 0.39, Egger's $P$ = 0.32; Fig F in S1 Fig). However, there was no statistically significant association of the risk of overall CAs in offspring of women with type 2 diabetes (RR = 1.31, 95% CI 0.80 to 2.15, $I^2$ = 98.2%, $P$ < 0.001; Fig L1 in S1 Fig).

## Exposure to PGDM and CHDs in offspring

A total of 18 studies reported on the association between maternal PGDM and CHDs in offspring [4,6,10–12,28–30,38,44,47,49,52,54,56,63,68,74]. Our results suggested that there is a statistically significant increase in risk of CHDs in offspring of women with PGDM (RR = 3.46, 95% CI 2.77 to 4.32, $I^2$ = 98.2%, $P$ < 0.001; Fig 4), with no evidence of publication bias (Begg's $P$ = 0.60, Egger's $P$ = 0.85; Fig E in S1 Fig). Similarly, maternal type 1 and type 2 diabetes were associated with increased risk of CHDs in offspring (type 1: RR = 3.75, 95% CI

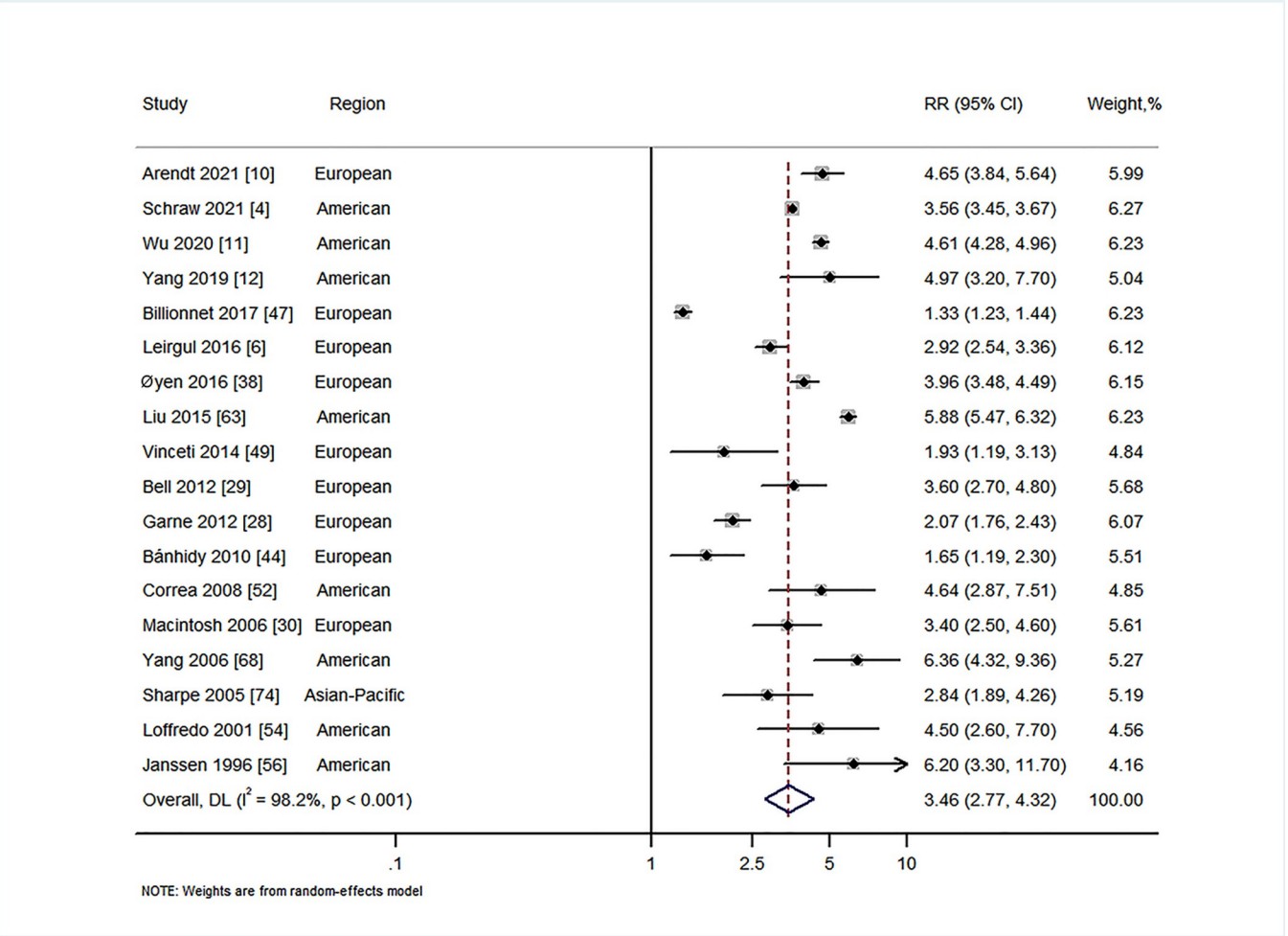

**Fig 4. Forest plot of the RRs in population-based studies formaternal pre-gestational diabetes and the risk of congenital heart defects (RR = 3.46, 95% CI 2.77 to 4.32, $I^2$ = 98.2%, $P$ < 0.001).** Analytical weights are from random effects meta-analysis. Grey boxes represent study estimates; their size is proportional to the respective analytical weight. Lines through the boxes represent the 95% CIs around the study estimates. The diamond represents the mean estimate and its 95% CI. The vertical red dashed line indicates the mean estimate. CI, confidence interval; DL, DerSimonian and Laird random effects model; RR, relative risk.

1.86 to 7.57, $I^2$ = 99.1%, $P$ < 0.001; Fig K2 in S1 Fig; type 2: RR = 3.15, 95% CI 1.72 to 5.78, $I^2$ = 93.6%, $P$ < 0.001; Fig J2 in S1 Fig). Notably, we found that maternal PGDM was associated with increased risk of all specific types of CHDs available for examination in the present study. The RRs of specific types of CHDs ranged from 2.23 (for hypoplastic left heart, 95% CI 1.07 to 4.64, $I^2$ = 64.0%, $P$ = 0.040) to 12.16 (for truncus arteriosus, 95% CI 7.52 to 19.68, $I^2$ = 0%, $P$ = 0.866) (Table 2; Figs G1–G18 in S1 Fig).

## Exposure to GDM and CHDs in offspring

Eleven studies explored the relationship between GDM and CHDs in offspring [4,6,11,12,38,44,47,51,52,56,72]. Our results suggested that maternal GDM is associated with CHDs (RR = 1.50, 95% CI 1.38 to 1.64, $I^2$ = 81.2%, $P$ < 0.001; Fig 5), with no evidence of publication bias (Begg's $P$ = 0.837, Egger's $P$ = 0.885; Fig F in S1 Fig). Regarding specific types of CHDs, we found that offspring of women with GDM had an increased risk of heterotaxia

**Table 2. Pooled RR and 95% confidence intervals for associations between maternal diabetes and any type of congenital heart defects .**

| Outcome | Number of events | Pre-gestational diabetes | | | | Gestational diabetes | | | |
|---|---|---|---|---|---|---|---|---|---|
| | | Number of studies | Pooled RR (95% CI) | $I^2$ (%) | P value | Number of studies | Pooled RR (95% CI) | $I^2$ (%) | P value |
| **Heterotaxia** | 1,098 | 4 | 8.78 (6.66 to 11.56) | 0.0 | 0.423 | 2 | 5.70 (1.09 to 29.92) | 85.7 | 0.008 |
| **Conotruncal defects** | 5,495 | 4 | 3.76 (2.58 to 5.48) | 68.3 | 0.024 | — | | | |
| Truncus arteriosus | 435 | 3 | 12.16 (7.52 to 19.68) | 0.0 | 0.866 | 2 | 1.77 (0.80 to 3.92) | 40.2 | 0.196 |
| Transposition of great vessels | 6,700 | 9 | 3.25 (2.54 to 4.15) | 15.9 | 0.301 | 2 | 1.29 (0.99 to 1.67) | 61.2 | 0.109 |
| Tetralogy of Fallot | 5,360 | 6 | 3.46 (2.27 to 5.28) | 64.4 | 0.015 | 2 | 1.41 (1.20 to 1.66) | 0.0 | 0.600 |
| **APVR** | 1,239 | 4 | 3.47 (2.13 to 5.64) | 0.0 | 0.684 | 2 | 1.42 (0.79 to 2.56) | 53.3 | 0.117 |
| **LVOT defects** | 6,672 | 7 | 3.46 (2.59 to 4.62) | 37.8 | 0.140 | 4 | 1.67 (1.15 to 2.41) | 50.0 | 0.112 |
| Coarctation of aorta | 6,606 | 5 | 3.35 (2.25 to 4.99) | 61.4 | 0.035 | 2 | 1.50 (1.23 to 1.83) | 35.4 | 0.213 |
| Hypoplastic left heart | 2,319 | 4 | 2.23 (1.07 to 4.64) | 64.0 | 0.040 | 2 | 1.23 (0.54 to 2.82) | 81.7 | 0.019 |
| **RVOT defects** | 6,163 | 7 | 3.41 (2.65 to 4.38) | 20.9 | 0.270 | 3 | 1.25 (1.03 to 1.53) | 0.0 | 0.739 |
| Pulmonary artery anomalies | 17,215 | 3 | 2.81 (2.48 to 3.18) | 0.0 | 0.865 | 2 | 1.02 (0.36 to 2.87) | 71.6 | 0.060 |
| Pulmonary valve stenosis | 7,273 | 5 | 2.51 (1.51 to 4.17) | 76.2 | 0.002 | 2 | 1.30 (0.96 to 1.76) | 64.5 | 0.093 |
| **Septal defects** | 12,368 | 2 | 3.23 (2.20 to 4.74) | 86.2 | 0.007 | — | | | |
| AVSD | 5,126 | 6 | 3.94 (2.95 to 5.26) | 40.0 | 0.139 | 3 | 1.02 (0.83 to 1.24) | 0.0 | 0.751 |
| VSD | 64,844 | 10 | 3.10 (2.32 to 4.16) | 90.2 | <0.001 | 2 | 1.31 (1.24 to 1.38) | 0.0 | 0.960 |
| ASD | 91,683 | 7 | 3.12 (2.42 to 4.02) | 81.9 | <0.001 | 2 | 1.45 (1.40 to 1.50) | 0.0 | 0.426 |
| VSD + ASD | 1,089 | 2 | 6.36 (4.38 to 9.24) | 0.0 | 0.527 | — | | | |
| **Single ventricle** | 1,228 | 4 | 5.91 (2.43 to 14.38) | 80.2 | 0.002 | 2 | 1.14 (0.77 to 1.69) | 0.0 | 0.851 |

APVR, anomalous pulmonary venous return; ASD, atrial septal defect; AVSD, atrioventricular septal defect; CHD, congenital heart defect; CI, confidence interval; LVOT, left ventricular outflow tract; RR, relative risk; RVOT, right ventricular outflow tract; VSD, ventricular septal defect.

(RR = 5.70, 95% CI 1.09 to 29.92, $I^2$ = 85.7%, $P$ = 0.008), tetralogy of Fallot (RR = 1.41, 95% CI 1.20 to 1.66, $I^2$ = 0%, $P$ = 0.600), left ventricular outflow tract defect (RR = 1.67, 95% CI 1.15 to 2.41, $I^2$ = 50.0%, $P$ = 0.112), coarctation of aorta (RR = 1.50, 95% CI 1.23 to 1.83, $I^2$ = 35.4%, $P$ = 0.213), right ventricular outflow tract defect (RR = 1.25, 95% CI 1.03 to 1.53, $I^2$ = 0%, $P$ = 0.739), ventricular septal defect (RR = 1.31, 95% CI 1.24 to 1.38, $I^2$ = 0%, $P$ = 0.960), and atrial septal defect (RR = 1.45, 95% CI 1.40 to 1.50, $I^2$ = 0%, $P$ = 0.426) (Table 2; Figs I1–I15 in S1 Fig).

## Exposure to PGDM and other type-specific CAs in offspring

We examined the associations between maternal PGDM and other type-specific CAs in offspring. Our results suggested that offspring of women with PGDM had an increased risk of CAs of the nervous system (RR = 2.54, 95% CI 1.73 to 3.73, $I^2$ = 94.8%, $P$ < 0.001); eye, ear, face, and neck (RR = 3.14, 95% CI 2.90 to 3.39, $I^2$ = 0%, $P$ = 0.444); digestive system (RR = 2.02, 95% CI 1.24 to 3.28, $I^2$ = 92.3%, $P$ < 0.001); genitourinary system (RR = 1.73, 95% CI 1.35 to 2.21, $I^2$ = 89.2%, $P$ < 0.001); and musculoskeletal system (RR = 1.98, 95% CI 1.45 to 2.72, $I^2$ = 94.4%, $P$ < 0.001), as well as an increased risk of multiple CAs (RR = 3.06, 95% CI 2.36 to 3.96, $I^2$ = 39.6%, $P$ = 0.158). The associations were statistically significant in 14 of 16 type-specific CA categories. The corresponding RRs ranged from 1.57 (for hypospadias, 95% CI 1.22 to 2.02, $I^2$ = 74.1%, $P$ < 0.001) to 18.18 (for holoprosencephaly, 95% CI 4.03 to 82.06, $I^2$ = 66.3%, $P$ = 0.085) (Table 3; Figs H1–H25 in S1 Fig).

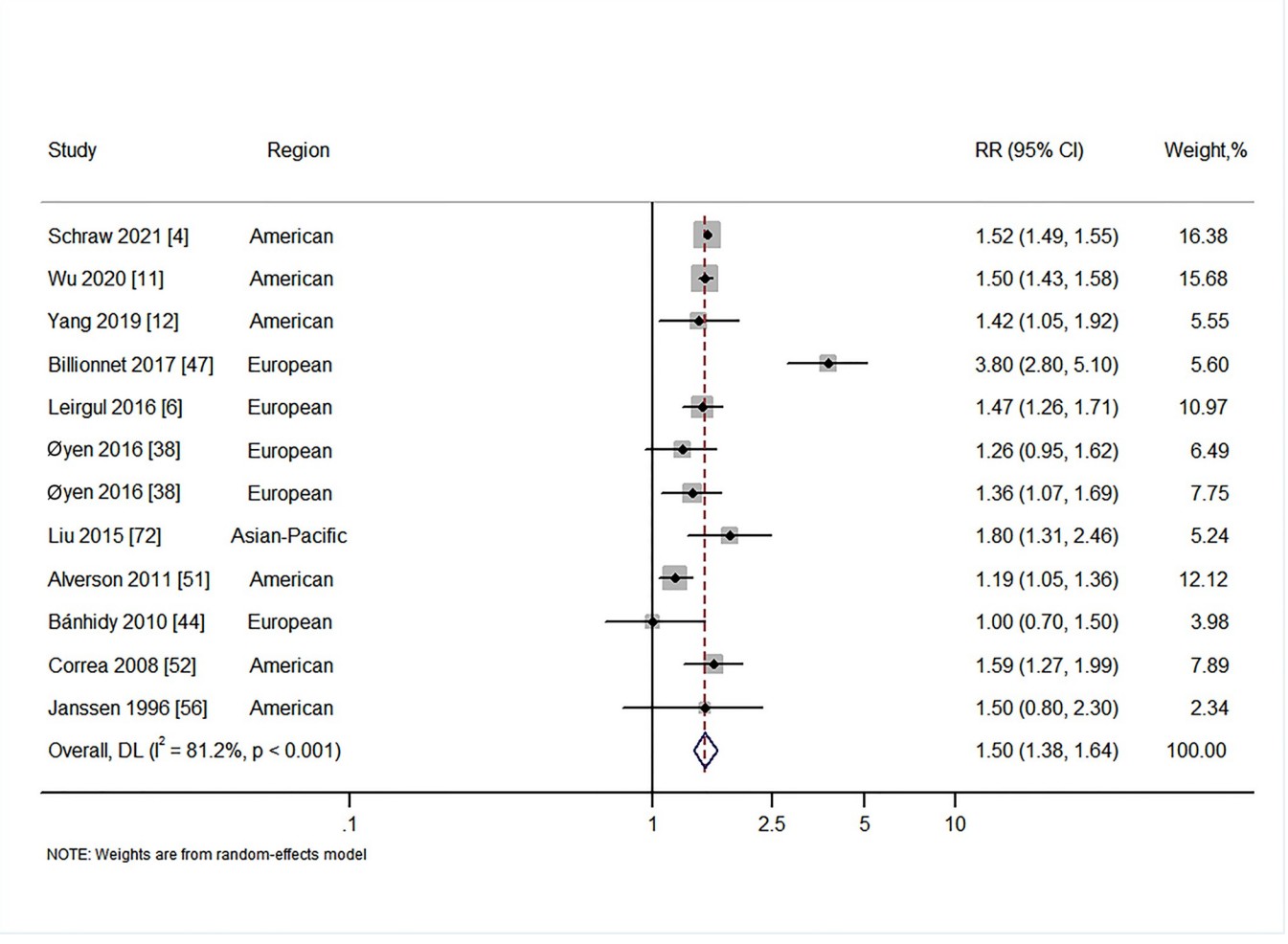

**Fig 5. Forest plot of the RRs in population-based studies for maternal gestational diabetes and the risk of congenital heart defects (RR = 1.50, 95% CI 1.38 to 1.64, $I^2$ = 81.2%, $P$ < 0.001).** Analytical weights are from random effects meta-analysis. Grey boxes represent study estimates; their size is proportional to the respective analytical weight. Lines through the boxes represent the 95% CIs around the study estimates. The diamond represents the mean estimate and its 95% CI. The vertical red dashed line indicates the mean estimate. CI, confidence interval; DL, DerSimonian and Laird random effects model; RR, relative risk.

## Exposure to GDM and other type-specific CAs in offspring

Maternal GDM was associated with an increased risk of CAs of the eye, ear, face, and neck (RR = 1.15, 95% CI 1.09 to 1.22, $I^2$ = 0%, $P$ = 0.355) and musculoskeletal system (RR = 1.18, 95% CI 1.15 to 1.22, $I^2$ = 0%, $P$ = 0.424) in offspring. In addition, maternal GDM also contributed to an increased risk of specific types of CAs in offspring, including hydrocephaly (RR = 1.34, 95% CI 1.16 to 1.54, $I^2$ = 0%, $P$ = 0.960), holoprosencephaly (RR = 1.87, 95% CI 1.09 to 3.22, $I^2$ = 0%, $P$ = 0.558), cleft lip with or without cleft palate (RR = 1.26, 95% CI 1.19 to 1.34, $I^2$ = 0%, $P$ = 0.547), diaphragmatic hernia (RR = 1.21, 95% CI 1.08 to 1.37, $I^2$ = 0%, $P$ = 0.779), omphalocele (RR = 1.21, 95% CI 1.05 to 1.40, $I^2$ = 0%, $P$ = 0.743), and hypospadias (RR = 1.29, 95% CI 1.16 to 1.44, $I^2$ = 45.9%, $P$ = 0.100) (Table 3; Figs J1–J22 in S1 Fig).

## Subgroup, meta-regression, and sensitivity analyses

The sensitivity analysis evaluated the effect of omitting 1 study at a time from each analysis. In the sensitivity analysis, we observed that the high $I^2$ value of 81.2% shown in the results for

**Table 3. Pooled RRs and 95% confidence intervals for associations between maternal diabetes and other type-specific congenital anomalies.**

| Outcome | Number of events | Pre-gestational diabetes | | | | Gestational diabetes | | | |
|---|---|---|---|---|---|---|---|---|---|
| | | Number of studies | Pooled RR (95% CI) | $I^2$ (%) | P value | Number of studies | Pooled RR (95% CI) | $I^2$ (%) | P value |
| **Nervous system defects** | 42,339 | 9 | 2.54 (1.73 to 3.73) | 94.8 | <0.001 | 2 | 1.64 (0.74 to 3.61) | 78.6 | 0.031 |
| Neural tube defects | 8,791 | 6 | 2.74 (1.46 to 5.14) | 75.5 | 0.001 | 2 | 1.06 (0.55 to 2.06) | 0.0 | 0.669 |
| Anencephaly | 3,859 | 3 | 2.72 (2.16 to 3.44) | 0.0 | 0.416 | 3 | 0.80 (0.62 to 1.04) | 25.4 | 0.262 |
| Encephalocele | 1,108 | 3 | 5.53 (3.24 to 9.45) | 52.8 | 0.120 | 2 | 1.03 (0.67 to 1.59) | 3.5 | 0.309 |
| Spina bifida | 9,948 | 8 | 1.89 (1.15 to 3.09) | 71.1 | 0.001 | 5 | 1.10 (0.99 to 1.22) | 0.0 | 0.459 |
| Hydrocephaly | 10,733 | 5 | 3.46 (1.62 to 7.42) | 85.0 | <0.001 | 4 | 1.34 (1.16 to 1.54) | 0.0 | 0.960 |
| Holoprosencephaly | 301 | 2 | 18.18 (4.03 to 82.06) | 66.3 | 0.085 | 3 | 1.87 (1.09 to 3.22) | 0.0 | 0.558 |
| **Eye, ear, face, and neck defects** | 39,570 | 6 | 3.14 (2.90 to 3.39) | 0.0 | 0.444 | 2 | 1.15 (1.09 to 1.22) | 0.0 | 0.355 |
| **Orofacial clefts** | 6,602 | 5 | 1.27 (0.54 to 2.98) | 90.4 | <0.001 | — | | | |
| Cleft palate | 11,259 | 6 | 1.75 (1.04 to 2.94) | 74.6 | 0.001 | 5 | 1.21 (0.95 to 1.56) | 54.9 | 0.064 |
| Cleft lip with or without cleft palate | 32,641 | 7 | 1.89 (1.22 to 2.92) | 81.1 | <0.001 | 5 | 1.26 (1.19 to 1.34) | 0.0 | 0.547 |
| **Digestive system defects** | 14,286 | 7 | 2.02 (1.24 to 3.28) | 92.3 | <0.001 | — | | | |
| Diaphragmatic hernia | 5,882 | 3 | 1.66 (1.32 to 2.10) | 0.0 | 0.520 | 4 | 1.21 (1.08 to 1.37) | 0.0 | 0.779 |
| **Abdominal wall defects** | 1,691 | 2 | 1.31 (0.80 to 2.15) | 0.0 | 0.729 | — | | | |
| Omphalocele | 4,163 | 3 | 1.90 (1.48 to 2.44) | 0.0 | 0.447 | 2 | 1.21 (1.05 to 1.40) | 0.0 | 0.743 |
| Gastroschisis | 9,268 | 3 | 0.92 (0.68 to 1.24) | 0.0 | 0.399 | 4 | 0.71 (0.58 to 0.85) | 0.0 | 0.424 |
| **Genitourinary system defects** | 128,657 | 10 | 1.73 (1.35 to 2.21) | 89.2 | <0.001 | 2 | 1.82 (0.90 to 3.66) | 93.4 | <0.001 |
| Renal agenesis/dysgenesis | 5,239 | 6 | 5.63 (2.48 to 12.76) | 86.1 | <0.001 | 2 | 0.90 (0.25 to 3.25) | 78.8 | 0.030 |
| Hypospadias | 44,963 | 9 | 1.57 (1.22 to 2.02) | 74.1 | <0.001 | 6 | 1.29 (1.16 to 1.44) | 45.9 | 0.100 |
| CAKUT | 4,143 | 3 | 1.80 (1.41 to 2.30) | 0.0 | 0.865 | 3 | 1.28 (0.99 to 1.66) | 31.1 | 0.234 |
| **Musculoskeletal system defects** | 123,365 | 11 | 1.98 (1.45 to 2.72) | 94.4 | <0.001 | 3 | 1.18 (1.15 to 1.22) | 0.0 | 0.424 |
| Limb reduction | 23,963 | 9 | 2.73 (1.98 to 3.76) | 81.7 | <0.001 | 5 | 1.14 (1.06 to 1.23) | 0.0 | 0.866 |
| Polydactyly/syndactyly | 20,328 | 4 | 0.95 (0.57 to 1.57) | 71.8 | 0.003 | 2 | 0.84 (0.42 to 1.66) | 62.5 | 0.102 |
| **Multiple congenital anomalies** | 2,448 | 5 | 3.06 (2.36 to 3.96) | 39.6 | 0.158 | 2 | 1.15 (0.59 to 2.24) | 63.0 | 0.100 |
| **Major congenital anomalies** | 52,171 | 6 | 2.14 (1.65 to 2.77) | 81.8 | <0.001 | 3 | 1.23 (1.03 to 1.47) | 18.5 | 0.293 |

CAKUT, congenital anomalies of the kidney and urinary tract; CI, confidence interval; RR, relative risk.

CHDs in offspring of mothers with GDM reduced to a moderate $I^2$ value of 55.0% when excluding the study by Billionnet et al. [47]. Although the increased risk association remained robust across scenarios, some moderate to significant heterogeneity still persisted and could not be reduced in sensitivity analyses. To explore the source of heterogeneity, we performed subgroup and meta-regression analyses in the predefined subgroups of study location, year of enrollment, study sample size, and adjustment for confounders (Tables 4 and 5). The findings of increased overall CA/CHD risk associated with maternal diabetes were consistently observed in most of the subgroup analyses. The results of the subgroup analyses suggested that differences in study sample size, population region, year of enrollment, and adjustment for confounders were major sources of heterogeneity. We observed that the high $I^2$ value of 81.2% observed in the results for CHDs in offspring of mothers with GDM was reduced to no ($I^2 = 0\%$) or low ($I^2 = 47.7\%$) heterogeneity after adjustment for race/ethnicity, body mass index, education, smoking/alcohol consumption, parity, and pregnancy complications (Table 5). In addition, the results of meta-regression analyses showed statistically significant differences in the RRs of CAs/CHDs in PGDM versus GDM (all $P_{\text{meta-regression}} < 0.001$) (Fig 6).

**Table 4. Subgroup analysis of the association between maternal diabetes and risk of overall congenital anomalies in offspring: Results of meta-analyses.**

| Subgroup | Pre-gestational diabetes | | | | | Gestational diabetes | | | | |
|---|---|---|---|---|---|---|---|---|---|---|
| | Number of studies | Pooled RR (95% CI) | $I^2$ (%) | P value* | P value** | Number of studies | Pooled RR (95% CI) | $I^2$ (%) | P value* | P value** |
| **Region** | | | | | 0.39 | | | | | 0.32 |
| Europe | 6 | 2.10 (1.44 to 3.04) | 94.5 | <0.001 | | 4 | 1.09 (0.97 to 1.23) | 73.7 | <0.001 | |
| North/South America | 10 | 2.03 (1.85 to 2.22) | 88.5 | <0.001 | | 10 | 1.19 (1.13 to 1.25) | 79.5 | <0.001 | |
| Asia-Pacific | 3 | 1.67 (1.31 to 2.11) | 42.1 | 0.178 | | 1 | 1.18 (0.79 to 1.77) | — | — | |
| **Year of enrollment**[†] | | | | | 0.87 | | | | | 0.18 |
| Before 1997 | 8 | 1.94 (1.51 to 2.49) | 93.2 | <0.001 | | 5 | 1.10 (0.97 to 1.25) | 81.1 | <0.001 | |
| In or after 1997 | 11 | 2.04 (1.86 to 2.23) | 90.0 | <0.001 | | 10 | 1.21 (1.16 to 1.26) | 67.4 | 0.001 | |
| **Number of participants**[†] | | | | | 0.44 | | | | | 0.78 |
| <282,260 | 4 | 1.56 (3.32 to 9.74) | 57.9 | 0.068 | | 3 | 1.40 (0.71 to 2.77) | 47.9 | 0.147 | |
| ≥282,260 | 15 | 1.96 (1.79 to 2.15) | 91.7 | <0.001 | | 12 | 1.17 (1.13 to 1.22) | 79.5 | <0.001 | |
| **Adjustment for confounders** | | | | | | | | | | |
| Maternal age | | | | | 0.19 | | | | | 0.71 |
| Yes | 16 | 1.92 (1.77 to 2.10) | 89.0 | <0.001 | | 13 | 1.17 (1.13 to 1.22) | 77.8 | <0.001 | |
| No | 3 | 2.31 (1.17 to 4.57) | 91.8 | <0.001 | | 2 | 2.59 (0.36 to 18.9) | 73.3 | 0.053 | |
| Race/ethnicity | | | | | 0.14 | | | | | 0.24 |
| Yes | 7 | 2.20 (1.98 to 2.45) | 88.6 | <0.001 | | 5 | 1.22 (1.17 to 1.27) | 74.2 | 0.004 | |
| No | 12 | 1.82 (1.54 to 2.16) | 91.2 | <0.001 | | 10 | 1.13 (1.04 to 1.23) | 73.3 | <0.001 | |
| Body mass index | | | | | 0.70 | | | | | 0.70 |
| Yes | 4 | 2.08 (1.55 to 2.80) | 83.4 | <0.001 | | 4 | 1.19 (1.08 to 1.32) | 74.3 | 0.009 | |
| No | 15 | 1.96 (1.78 to 2.17) | 89.6 | <0.001 | | 11 | 1.16 (1.11 to 1.22) | 68.8 | <0.001 | |
| Education | | | | | 0.40 | | | | | 0.76 |
| Yes | 5 | 1.85 (1.59 to 2.15) | 93.7 | <0.001 | | 6 | 1.17 (1.10 to 1.24) | 87.7 | <0.001 | |
| No | 14 | 2.05 (1.80 to 2.33) | 88.0 | <0.001 | | 9 | 1.18 (1.12 to 1.26) | 33.9 | 0.147 | |
| Smoking/alcohol consumption | | | | | 0.55 | | | | | 0.49 |
| Yes | 6 | 2.11 (1.67 to 2.66) | 79.1 | <0.001 | | 6 | 1.21 (1.11 to 1.32) | 59.7 | 0.030 | |
| No | 13 | 1.94 (1.75 to 2.15) | 90.5 | <0.001 | | 9 | 1.16 (1.10 to 1.21) | 73.4 | <0.001 | |
| Parity | | | | | 0.89 | | | | | 0.92 |
| Yes | 8 | 1.99 (1.80 to 2.20) | 90.1 | <0.001 | | 8 | 1.17 (1.11 to 1.23) | 84.4 | <0.001 | |
| No | 11 | 1.98 (1.59 to 2.47) | 90.8 | <0.001 | | 7 | 1.18 (1.08 to 1.29) | 43.0 | 0.104 | |
| Pregnancy complications | | | | | 0.05 | | | | | 0.44 |
| Yes | 6 | 1.69 (1.29 to 2.20) | 93.6 | <0.001 | | 6 | 1.13 (1.02 to 1.25) | 87.5 | <0.001 | |
| No | 13 | 2.20 (2.00 to 2.42) | 90.0 | <0.001 | | 9 | 1.20 (1.16 to 1.24) | 28.4 | 0.192 | |

CI, confidence interval; RR, relative risk.

*P for heterogeneity within each subgroup.

**P for heterogeneity between subgroups with meta-regression analysis.

[†]Categorized using the median as the cutoff value.

## Discussion

To the best of our knowledge, the present study is the first comprehensive systematic review and meta-analysis of population-based studies of over 80 million participants that shows an increased risk of type-specific CAs, especially CHDs, in offspring of women with pre-gestational or gestational diabetes. The study findings suggested that maternal PGDM was associated with a significant increase in the risk of CAs in offspring in 38 of 45 categories, while

**Table 5. Subgroup analysis of the association between maternal diabetes and risk of congenital heart defects in offspring: Results of meta-analyses.**

| Subgroup | Pre-gestational diabetes | | | | | Gestational diabetes | | | | |
|---|---|---|---|---|---|---|---|---|---|---|
| | Number of studies | Pooled RR (95% CI) | $I^2$ (%) | P value* | P value** | Number of studies | Pooled RR (95% CI) | $I^2$ (%) | P value* | P value** |
| **Region** | | | | | **0.04** | | | | | 0.98 |
| Europe | 9 | 2.63 (1.81 to 3.80) | 97.6 | <0.001 | | 4 | 1.57 (1.09 to 2.27) | 90.8 | <0.001 | |
| North/South America | 8 | 4.90 (3.92 to 6.13) | 96.2 | <0.001 | | 6 | 1.46 (1.37 to 1.56) | 64.0 | 0.016 | |
| Asia-Pacific | 1 | 2.84 (1.89 to 4.26) | — | — | | 1 | 1.80 (1.31 to 1.55) | — | — | |
| **Year of enrollment†** | | | | | 0.46 | | | | | 0.07 |
| Before 1997 | 10 | 3.17 (2.49 to 4.03) | 88.4 | <0.001 | | 5 | 1.29 (1.17 to 1.44) | 23.9 | 0.255 | |
| In or after 1997 | 8 | 3.80 (2.67 to 5.40) | 99.2 | <0.001 | | 6 | 1.68 (1.50 to 1.88) | 86.7 | <0.001 | |
| **Number of participants†** | | | | | 0.20 | | | | | 0.62 |
| <282,260 | 3 | 4.93 (3.61 to 6.74) | 0.0 | 0.714 | | 3 | 1.37 (1.09 to 1.72) | 61.1 | 0.08 | |
| ≥282,260 | 15 | 3.27 (2.57 to 4.15) | 98.5 | <0.001 | | 8 | 1.55 (1.40 to 1.70) | 82.3 | <0.001 | |
| **Adjustment for confounders** | | | | | | | | | | |
| Maternal age | | | | | 0.97 | | | | | 0.40 |
| Yes | 16 | 3.46 (2.72 to 4.39) | 98.4 | <0.001 | | 10 | 1.55 (1.42 to 1.69) | 77.9 | <0.001 | |
| No | 2 | 3.50 (2.84 to 4.32) | 0.0 | 0.789 | | 1 | 1.19 (1.05 to 1.35) | — | — | |
| Race/ethnicity | | | | | 0.21 | | | | | 0.94 |
| Yes | 6 | 4.14 (3.42 to 5.01) | 89.3 | <0.001 | | 5 | 1.52 (1.49 to 1.55) | 0.0 | 0.964 | |
| No | 12 | 3.15 (2.11 to 4.69) | 98.7 | <0.001 | | 6 | 1.53 (1.19 to 1.98) | 89.1 | <0.001 | |
| Body mass index | | | | | 0.17 | | | | | 0.83 |
| Yes | 4 | 4.62 (4.30 to 4.96) | 0.0 | 0.989 | | 4 | 1.51 (1.44 to 1.58) | 0.0 | 0.654 | |
| No | 14 | 3.20 (2.45 to 4.20) | 98.5 | <0.001 | | 7 | 1.50 (1.26 to 1.77) | 87.7 | <0.001 | |
| Education | | | | | 0.38 | | | | | 0.90 |
| Yes | 3 | 4.18 (3.32 to 5.27) | 95.2 | <0.001 | | 4 | 1.52 (1.49 to 1.55) | 0.0 | 0.671 | |
| No | 15 | 3.33 (2.34 to 4.75) | 98.3 | <0.001 | | 7 | 1.51 (1.20 to 1.89) | 87.0 | <0.001 | |
| Smoking/alcohol consumption | | | | | **0.02** | | | | | 0.91 |
| Yes | 4 | 4.68 (4.38 to 5.01) | 0.0 | 0.491 | | 4 | 1.50 (1.43 to 1.58) | 0.0 | 0.706 | |
| No | 14 | 2.98 (2.24 to 3.96) | 98.6 | <0.001 | | 7 | 1.50 (1.29 to 1.76) | 87.8 | <0.001 | |
| Parity | | | | | 0.17 | | | | | 0.86 |
| Yes | 6 | 4.23 (3.38 to 5.30) | 97.5 | <0.001 | | 4 | 1.52 (1.49 to 1.55) | 0.0 | 0.676 | |
| No | 12 | 3.13 (2.20 to 4.47) | 96.6 | <0.001 | | 7 | 1.50 (1.17 to 1.93) | 86.9 | <0.001 | |
| Pregnancy complications | | | | | 0.88 | | | | | 0.61 |
| Yes | 4 | 3.58 (2.11 to 6.09) | 91.6 | <0.001 | | 4 | 1.45 (1.23 to 1.71) | 47.7 | 0.13 | |
| No | 14 | 3.44 (2.62 to 4.50) | 98.5 | <0.001 | | 7 | 1.56 (1.33 to 1.82) | 86.7 | <0.001 | |

CI, confidence interval; RR, relative risk.

*P for heterogeneity within each subgroup.

**P for heterogeneity between subgroups with meta-regression analysis.

†Categorized using the median as the cutoff value.

maternal GDM was associated with a small but significant increase in the risk of CAs in 16 of 45 categories. The corresponding RRs of overall CAs/CHDs in offspring of women with PGDM were higher than those in offspring of women with GDM, with no overlap in the 95% CIs.

Although the exact pathophysiology of the relationship between maternal diabetes and CAs in offspring remains unclear, metabolic changes in women with diabetes could play a critical

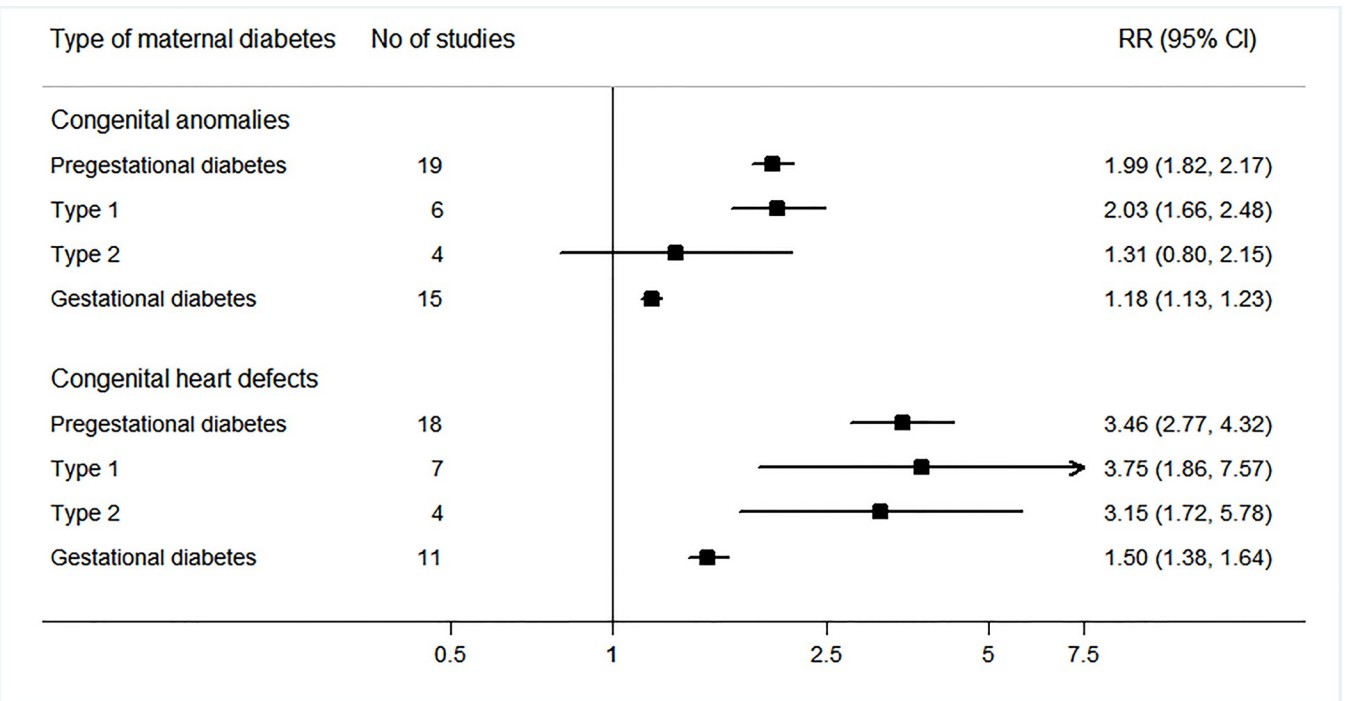

**Fig 6. Risks of overall congenital anomalies and congenital heart defects in offspring according to different types of maternal diabetes.** Relative risks (RRs) and 95% confidence intervals (CIs) are presented to show the risk of overall congenital anomalies and congenital heart defects in offspring born to women with different types of maternal diabetes compared with the risk among offspring born to women without diabetes. $P_{\text{meta-regression}}$ values were <0.001 for the comparison within congenital anomalies and congenital heart defects between gestational diabetes and pre-gestational diabetes.

role in the development of CAs in their offspring. Sustained hyperglycemia is the main characteristic of diabetes; this activates multiple metabolic pathways that play a role in the formation of CAs [77]. Notably, a common mechanism behind diabetic complications is mitochondrial superoxide radical production [77]. The production of reactive oxygen species (ROS) is induced by hyperglycemia, which is the crucial process in diabetes mellitus pathogenesis, and oxidative stress (OS) is known to affect embryonic development [77–79]. Although OS does not cause a direct genotoxic effect, a previous study showed that OS affects the expression of some genes involved in the various stages of embryonic development, and each gene affected might have a specific sensitivity to hyperglycemic conditions and changes in the cellular redox state, thus mediating the formation of CAs in offspring [77]. Studies have suggested that the formation of CAs involves AMP-activated protein kinase (AMPK), an enzyme with kinase activity that is activated in response to an increase in adenosine monophosphate nucleoside levels [78,80]. AMPK could play a key role in the formation of CAs; it participates in the regulation of energy metabolism and, once activated, moves into the cell nucleus and phosphorylates multiple proteins, including hypoxia-inducible factor 1α, which could mediate the development of CAs [77]. However, further study is needed to discern the exact mechanisms involved in the activation of AMPK and the induction of CAs by ROS.

An important finding from our meta-regression analyses is the statistically significant difference in the risk of overall CAs in offspring of women with PGDM versus offspring of women with GDM. That is, the risk of CAs in offspring was higher in women with PGDM than in those with GDM. Pregnancy begins at fertilization, and organogenesis begins during the third to eighth week post-conception and continues until birth. Therefore, the first trimester of pregnancy is the most critical period for organogenesis. In women with PGDM, there

can be a lengthy period of sustained hyperglycemia before and during pregnancy, which can significantly impact organogenesis and contribute to CAs in offspring. This differs from GDM, which is usually diagnosed during the 24th to 28th week of gestation [8]. Therefore, in a woman with GDM, blood glucose levels could be normal or just slightly elevated during the first trimester, leading to minimal influence on organogenesis. This could partly explain why offspring of women with PGDM were at greater risk for CAs compared with offspring of women with GDM. However, women who develop GDM during pregnancy usually have evidence of metabolic dysfunction before pregnancy, such as pancreatic β-cell defects and increased insulin resistance [81,82], which may contribute to the development of hyperglycemia and thus increase the rate of malformations in infants, although further studies are needed to elucidate the potential mechanisms involved. Another key finding from our meta-regression analysis was the similar result observed regarding CHDs in offspring of women with PGDM and offspring of women with GDM. The heart is the first functional organ to develop and starts to beat and pump blood at around 22 days after fertilization [83]. The septum, including the interatrial septum, begins to form at 4 to 7 weeks of gestation.

Hyperglycemia could have a more critical influence on heart development in the early stage of pregnancy than in the late stage of pregnancy. Therefore, screening for diabetes in pregnant women will enable better glycemic control, which might reduce the rate of malformations, especially during organogenesis. However, the exact mechanisms underlying the influence of diabetes on organogenesis in different stages remain unclear and require further study.

In the present study, the results regarding specific types of CHDs in offspring with maternal PGDM were consistent with 2 previous meta-analyses [13,14]. A recent systematic review and meta-analysis conducted by Chen and colleagues involved a pooled analysis of 24 population-based studies and 18 hospital-based studies; the findings suggested that maternal GDM was significantly associated with the risk of most phenotypes of CHDs [13]. New data from population-based studies of more than 36 million births provided solid estimates of the associations between maternal GDM and specific types of CHDs in offspring [4,10–12]. However, these studies mainly focused on the association between different types of maternal diabetes and CHDs. Little is known about the association between maternal diabetes and other specific types of CAs in offspring or the extent to which types of maternal diabetes are associated with the increased risk of CAs.

One recent meta-analysis by Parimi and colleagues explored the association between maternal diabetes and the risk in offspring of CAKUT, which refers to a range of structural and functional anomalies of the kidney, collecting system, bladder, and urethra [84]. Our findings were in line with the results from Parimi et al. [84] that showed that offspring of mothers with PGDM had an almost 2-fold increased risk of CAKUT; however, results regarding the association between maternal GDM and the risk of CAKUT were inconsistent. Our findings demonstrated associations between maternal diabetes and 23 CA categories (excluding CHD-related categories) in offspring and suggested that offspring of women with PGDM had an increased risk of 21 specific types of CAs, while increased risks of 9 specific types of CAs were observed in offspring of women with GDM.

## Strengths and limitations

Our study has several strengths. The first strength is the large sample size of over 80 million births from population-based data, which provides robust evidence regarding the risk of CAs in offspring of women with diabetes and are widely generalizable. Second, our study examined the associations between maternal diabetes and various types of CAs across multiple categories of maternal diabetes. Unlike previous studies [13,14,84] that only assessed the risk of CHDs or

CAKUT in maternal diabetes, the present study systematically and quantitatively summarized the associations between maternal diabetes and 45 type-specific CAs in offspring. Third, consistent results of the pooled RRs supported the robustness of the findings of our study. Finally, the current study examined the extent to which types of maternal diabetes (i.e., pre-gestational and gestational) are associated with increased risk of CAs in offspring. The relative consistency of associations observed appears to support the hypothesis that maternal diabetes, especially PGDM, increases the likelihood of type-specific CAs in offspring.

However, several limitations should be noted. Although the increased risk association remained robust across various scenarios, some high levels of statistical heterogeneity generally persisted and could not be reduced in subgroup and sensitivity analyses. There were some causes of heterogeneity in the included studies. First, there is lack of consensus and uniformity in the screening standards and diagnostic criteria for GDM. Also, pre-pregnancy diabetes is sometimes unrecognized and discovered only during pregnancy as GDM, which could lead to overestimation of RRs associated with GDM. Second, the ascertainment of some CAs may vary substantially between studies. Some CAs are easy to ascertain (e.g., anencephaly), while some may not be recognized immediately after birth and may be discovered only later in infancy (e.g., milder atrial septal defects). This also contributes to the heterogeneity of the results. Third, most studies included live births only; the lack of information on stillbirths and terminations of pregnancy for fetal anomaly could introduce selection bias and lead to underestimation of the strength of the associations between maternal diabetes and risk of CAs in offspring. Fourth, there may be other unmeasured confounding factors in addition to those adjusted for in each study. In this regard, further study could be performed to reduce the aforementioned causes of heterogeneity in a more in-depth analysis. An additional limitation was that although we summarized and quantified the existing population-based data on overall CAs/CHD observed under maternal type 1 or type 2 diabetes, data on other type-specific CAs in offspring associated with maternal type 1 or type 2 diabetes are limited. Additional studies are needed to address this issue. Furthermore, information on treatment (e.g., insulin use) or how well-controlled blood glucose levels were in the study participants was not available in most of the studies included in the current study. Further work should strive to address this lack of information. Finally, we observed a negative association between maternal GDM and risk of gastroschisis. The reasons for why maternal GDM was inversely associated with the risk of gastroschisis are currently unknown; this finding warrants confirmation and further investigation in future studies. Residual confounding may contribute to the inverse association, but further confirmation is still needed.

## Conclusion

In the present study, we observed an increased rate of CAs in the offspring of women with maternal diabetes and noted the differences between PGDM and GDM. Considering the substantial rise in the prevalence of maternal diabetes over recent decades, the expectation that this prevalence will continue to increase, the number of pregnancies worldwide, and the significant individual and global burdens associated with CAs in offspring, screening for diabetes in pregnant women may enable better glycemic control, and may enable identification of offspring at risk for CAs.

## Supporting information

**S1 Fig. Risk of bias, funnel plots, and forest plots regarding the associations between maternal diabetes and congenital anomalies in offspring.** Fig A: Risk of bias summary: Effect on congenital anomalies in offspring of women with pre-gestational diabetes. Fig B:

Risk of bias summary: Effect on congenital heart defects in offspring of women with pre-gestational diabetes. Fig C: Risk of bias summary: The effect on congenital anomalies in offspring of women with gestational diabetes. Fig D: Risk of bias summary: The effect on congenital heart defects in offspring of women with gestational diabetes. Fig E: Funnel plots of the relative risks in population-based studies for pre-gestational diabetes mellitus and the risk of congenital anomalies. Fig F: Funnel plots of the relative risks in population-based studies for gestational diabetes mellitus and the risk of congenital anomalies. Fig G: Forest plot of the relative risks in population-based studies for maternal pre-gestational diabetes and the risk of any type of congenital heart defect—G1: heterotaxia; G2: conotruncal defects; G3: truncus arteriosus; G4: transposition of great vessels; G5: tetralogy of Fallot; G6: atrioventricular septal defects; G7: anomalous pulmonary venous return; G8: left ventricular outflow tract defect; G9: coarctation of aorta; G10: hypoplastic left heart; G11: right ventricular outflow tract defect; G12: pulmonary artery anomalies; G13: pulmonary valve stenosis; G14: septal defects; G15: ventricular septal defects; G16: atrial septal defects; G17: ventricular septal defect and atrial septal defect; G18: single ventricle. Fig H: Forest plot of the relative risks in population-based studies for maternal pre-gestational diabetes and the risk of other type-specific congenital anomalies— H1: nervous system defects; H2: neural tube defects; H3: anencephaly; H4: encephalocele; H5: spina bifida; H6: hydrocephaly; H7: holoprosencephaly; H8: eye, ear, face, and neck defects; H9: orofacial clefts; H10: cleft palate. H11: cleft lip with or without cleft palate; H12: digestive system defects; H13: diaphragmatic hernia; H14: abdominal wall defects; H15: omphalocele; H16: gastroschisis; H17: genitourinary system defects; H18: renal agenesis/dysgenesis; H19: hypospadias; H20: congenital anomalies of the kidney and urinary tract; H21: musculoskeletal system defects; H22: limb reduction; H23: polydactyly/syndactyly; H24: multiple congenital anomalies; H25: major congenital anomalies. Fig I: Forest plot of the relative risks in population-based studies for maternal gestational diabetes and the risk of any type of congenital heart defect—I1: heterotaxia; I2: truncus arteriosus; I3: transposition of great vessels; I4: tetralogy of Fallot; I5: atrioventricular septal defects; I6: anomalous pulmonary venous return; I7: left ventricular outflow tract defect; I8: coarctation of aorta; I9: hypoplastic left heart; I10: right ventricular outflow tract defect; I11: pulmonary artery anomalies; I12: pulmonary valve stenosis; I13: ventricular septal defects; I14: atrial septal defects; I15: single ventricle. Fig J: Forest plot of the relative risks in population-based studies for maternal gestational diabetes and the risk of other type-specific congenital anomalies—J1: nervous system; J2: neural tube defects; J3: anencephaly; J4: encephalocele; J5: spina bifida; J6: hydrocephaly; J7: holoprosencephaly; J8: eye, ear, face, and neck defects; J9: cleft palate; J10: cleft lip with or without cleft palate; J11: diaphragmatic hernia; J12: omphalocele; J13: gastroschisis; J14: genitourinary system defects; J15: renal agenesis/dysgenesis; J16: hypospadias; J17: congenital anomalies of the kidney and urinary tract; J18: musculoskeletal system defects; J19: limb reduction; J20: polydactyly/syndactyly; J21: multiple congenital anomalies; J22: major congenital anomalies. Fig K1: Forest plot of the relative risks in population-based studies for maternal type 1 diabetes and the risk of overall congenital anomalies. Fig K2: Forest plot of the relative risks in population-based studies for maternal type 1 diabetes and the risk of congenital heart defects. Fig L1: Forest plot of the relative risks in population-based studies for maternal type 2 diabetes and the risk of overall congenital anomalies. Fig L2: Forest plot of the relative risks in population-based studies for maternal type 2 diabetes and the risk of congenital heart defects.
(PDF)

**S1 Protocol. The registered protocol for this review in PROSPERO.**
(PDF)

**S1 Table. EUROCAT, ICD-10, and ICD-9 codes used to identify and define congenital anomalies.**
(DOCX)

**S2 Table. References of studies excluded in the systematic review and meta-analysis of population-based studies.**
(DOCX)

**S3 Table. Ascertainment of maternal diabetes of the included studies in the systematic review and meta-analysis of population-based studies.**
(DOCX)

**S4 Table. Characteristics of population-based studies of maternal diabetes and congenital anomalies.**
(DOCX)

**S1 Text. Search strategy.**
(DOCX)

**S2 Text. MOOSE checklist.**
(DOCX)

**S3 Text. PRISMA 2020 checklist.**
(DOCX)

## Acknowledgments

We thank Professor Qi-Jun Wu, who kindly provided suggestions for this meta-analysis.

## Author Contributions

**Conceptualization:** Tie-Ning Zhang, Xin-Yi Zhao, Ri Wen, Shan-Yan Gao.

**Data curation:** Tie-Ning Zhang, Xin-Yi Zhao, Wei Wang, Ri Wen, Shan-Yan Gao.

**Formal analysis:** Xin-Mei Huang, Xin-Yi Zhao, Wei Wang, Ri Wen, Shan-Yan Gao.

**Funding acquisition:** Wei Wang, Ri Wen, Shan-Yan Gao.

**Investigation:** Xin-Mei Huang, Wei Wang, Ri Wen, Shan-Yan Gao.

**Methodology:** Xin-Mei Huang, Ri Wen, Shan-Yan Gao.

**Project administration:** Xin-Mei Huang, Shan-Yan Gao.

**Resources:** Xin-Mei Huang, Shan-Yan Gao.

**Software:** Xin-Mei Huang, Wei Wang, Shan-Yan Gao.

**Supervision:** Shan-Yan Gao.

**Validation:** Xin-Yi Zhao, Shan-Yan Gao.

**Visualization:** Xin-Yi Zhao, Shan-Yan Gao.

**Writing – original draft:** Tie-Ning Zhang, Xin-Yi Zhao, Wei Wang, Shan-Yan Gao.

**Writing – review & editing:** Shan-Yan Gao.

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
