## [Editor Report · Decision Letter 0]

5 Apr 2021

Dear Dr Gao, 

Thank you for submitting your manuscript entitled "Risks of specific congenital anomalies in offspring of diabetic mothers: a systematic review and meta-analysis of population-based studies of more than 76 million births" for consideration by PLOS Medicine.

Your manuscript has now been evaluated by the PLOS Medicine editorial staff and I am writing to let you know that we would like to send your submission out for external peer review.

Kind regards,

Caitlin Moyer, Ph.D.

Associate Editor

PLOS Medicine

---

## [Decision Letter · Decision Letter 1]

13 Oct 2021

Dear Dr. Gao,

Thank you very much for submitting your manuscript "Risks of specific congenital anomalies in offspring of diabetic mothers: a systematic review and meta-analysis of population-based studies of more than 76 million births" (PMEDICINE-D-21-01554R1) for consideration at PLOS Medicine. 

Your paper was evaluated by a senior editor and discussed among all the editors here. It was also discussed with an academic editor with relevant expertise, and sent to four independent reviewers, including a statistical reviewer. The reviews are appended at the bottom of this email and any accompanying reviewer attachments can be seen via the link below:

[LINK]

In light of these reviews, I am afraid that we will not be able to accept the manuscript for publication in the journal in its current form, but we would like to consider a revised version that addresses the reviewers' and editors' comments. Obviously we cannot make any decision about publication until we have seen the revised manuscript and your response, and we plan to seek re-review by one or more of the reviewers. 

We expect to receive your revised manuscript by Nov 03 2021 11:59PM. Please email us (plosmedicine@plos.org) if you have any questions or concerns.

We look forward to receiving your revised manuscript. 

Sincerely,

Caitlin Moyer, Ph.D.

Associate Editor

PLOS Medicine

plosmedicine.org

1. Data availability statement: PLOS Medicine requires that the de-identified data underlying the specific results in a published article be made available, without restrictions on access, in a public repository or as Supporting Information at the time of article publication, provided it is legal and ethical to do so. Please see the policy at

http://journals.plos.org/plosmedicine/s/data-availability

and FAQs at

http://journals.plos.org/plosmedicine/s/data-availability#loc-faqs-for-data-policy

It seems as if the data underlying the analyses are provided in the metadata files provided, however, there do not seem to be any descriptions of these files.

2. Author Summary: Please move the Author Summary to follow the Abstract.

3. Author Summary (and throughout): Please temper statements of primacy with “To the best of our knowledge, this is the first comprehensive systematic review…” or similar.

4. Author Summary (and throughout): Please define all abbreviations at first use in the text (RR).

5. Abstract: Please note the eligibility criteria, including language restrictions, of included studies.

6. Abstract: Methods and Findings: Please quantify the main results (with 95% CIs and p values).

7. Abstract: Methods and Findings: In the last sentence of the Abstract Methods and Findings section, please describe the main limitation(s) of the study's methodology.

8. Abstract: Conclusion: Please address the study implications without overreaching what can be concluded from the data; the phrase "In this study, we observed ..." may be useful.

9. Introduction: Lines 121-124: Here and throughout, please revise to avoid causal language and please refer to associations instead: “...examined the extent to which types of maternal diabetes (i.e. pregestational or gestational) are associated with increased risk of CAs in offspring.” or similar.

10. Methods: Please update your search to the present time (currently search has an end date of December 2020). 

11. Methods: Line 154: Please reference the PRISMA flowchart.

12. Methods: Line 208-209: Please provide more detail on how adequacy of controlling for confounders was determined.

13. Results: For the main results presented in the tables and in the text, please report both 95% CIs and p values.

14. Results: Line 285-288: Please avoid language implying causality, and please refer to associations here: “...statistically significant effect of maternal type 1 and type 2 diabetes on CHD…” and “Notably, we found that maternal PGDM increased the risk of all available specific types of CHD…”

15. Discussion: Please present and organize the Discussion as follows: a short, clear summary of the article's findings; what the study adds to existing research and where and why the results may differ from previous research; strengths and limitations of the study; implications and next steps for research, clinical practice, and/or public policy; one-paragraph conclusion.

16. Discussion: Line 450-452: Here and throughout the discussion, please avoid the use of language implying causality: “Finally, the current study examined the extent to which types of maternal diabetes (i.e., pregestational or gestational) increase risk of CAs in their offspring.”

17. S1 Text: Thank you for including the PRISMA and MOOSE checklists. When completing the checklists, please use section and paragraph numbers rather than page numbers to refer to locations within the text.

Comments from the reviewers:

Reviewer #1: This systematic review and meta-analysis examines the association between pre-pregnancy diabetes mellitus, gestational diabetes and congenital anomalies. This is a thorough review, well written and comprehensively presented. I have only relatively minor comments.

Comments:

1. Table 1: I suppose type 1 and type 2 diabetes mellitus would always be 'pre-gestational diabetes'. In the category 'pre-gestational diabetes', did the results of all 42 studies relate to a combined any type 1 or type 2 diabetes? Perhaps, instead of 'pre-gestational diabetes', you can use, for instance, 'pre-gestational diabetes (combined type 1 or 2)'. Just a suggestion.

2. Similar issue in Table 2: The indentation of subcategories in the table suggests these sub-categories are included in the total above. Can you, please, clarify, for instance, 'LVOT' category includes all LVOT anomalies, perhaps instead of 'LVOT' you can use 'LVOT combined' or 'All LVOT). The same applies to 'Conotruncal defect' - you can use 'Conotruncal defects (any)', etc. Similar issue is in Table 3, most rows look like subcategories. Just a suggestion.

3. Page 11, Line 196: Do you mean effect size (instead of effective size)?

4. Page 11, line 209: Can you specify adequate and not adequate control for confounding? What set of confounders would you consider adequate adjustment in the studies that adjusted for confounders? They are listed in Supplemental tables (Table D) but what was considered an 'adequate control' is not stated.

5. Fig 1 suggests that after identification of 23,748 records in PubMed and Embase and 1 record extra, the majority was excluded due to duplicates ('records after duplicates removed n=2,096'). You could perhaps place this box with the right side arrow and use the text 'Duplicates removed (n=2,096)' 

6. In sensitivity analyses, strata categories are: before year 2000, between 2000 and 2010, and after 2010. It is not wrong, I am just wondering if there was any rationale for these categories.

7. Discussion: In study limitations, it would be nice to address the lack of information on how well the diabetes was controlled (e.g., mothers with well controlled diabetes would be expected to have lower rates of CA). Similarly, information on treatment (e.g., insulin use) was not available in most studies included in the review.

8. In Table 3, a relative risk of gastroschisis is below 1 (i.e., a 'protective effect' of maternal gestational diabetes). Is this consistent across the studies? Why would there be a negative association? It would be nice to mention in the discussion. 

9. You mentioned ascertainment of maternal diabetes in the limitations. Besides varying criteria for GDM, pre-pregnancy diabetes is sometimes unrecognized and discovered only during pregnancy as GDM, which could lead to overestimation of relative risks associated with GDM.

10. In addition to the limitations, the ascertainment of some congenital anomalies may vary substantially between studies. Some CAs are easy to ascertain (e.g., anencephaly), while some may not be recognized immediately after birth and be discovered only later in infancy (e.g., milder atrial septal defects). This also contributes to the heterogeneity of the results.

11. The manuscript needs some editing.

Minor comments:

Page 16 Line 287: "Notably, we found that maternal PGDM increased the risk of all available specific types of CHD in the present study". I suggest "Notably, we found that maternal PGDM increased the risk of all specific types of CHD available to examine in the present study".

Reviewer #2: This is a well-conducted systematic review and meta-analysis on the risks of specific congenital anomalies in offspring of diabetic mothers using population-based studies. The study design, datasets, statistical methods and analyses, and presentation (tables and figures) and interpretation of the results are mostly adequate and of a good standard. Only a few minor issues needing attention.

1) In table 1, For Type of maternal diabetes, the number of studies doesn't add to the 56 total by whatever combination of Pregestational diabetes, Type 1 diabetes, Type 2 diabetes, and Gestational diabetes. There must be overalps. Can authors make it clearer? especially what is the exact overlap and in which subtype?

2) Meta-regression analyses were nicely done to show significant differences in RRs of CAs/CHD in PGDM versus GDM. However, as the selected studies are over about 50 years' period, could the meta-analysis be done over time to show whether any difference in RRs of CAs/CHD over time?

3) On page 15, High I-squared of 90% was shown on the results for overall CAs in offspring of mothers with PGDM. Similar high I-squared were also found in the results with T1D (82.5%) and GDM (76%). This happened even after using the random effect model. Can authors explain this high heterogeneity and potential impact on the results?

Reviewer #3: Overview

This is a very extensive systematic review of the risks of specific congenital anomalies and GDM and PGDM. The authors have done an excellent job in collating all this information

Major Comments 

1. Table 3 - what is the bottom line based on 4 and 2 studies ? How does it relate to data in Figure E

2. Lines 327 - I do not understand what is being done here in the sensitivity analyses- the authors say that they reduced the I2 by excluding individual studies, with one range going from 55% to 82.9%. Does this not imply that one single study was responsible for a large amount of the variation and in which case more information should be given about that individual study and why it was so discrepant from the other studies.

3. Lines 389 - The authors state "Therefore, blood glucose during the first trimester in a mother with GDM could be normal or just slightly elevated, leading to minimal influence on organogenesis." And yet they present results showing that there is an increased risk of anomalies in mothers with GDM. This needs more clarification as to why you would expect an increased risk in GDM

4. The paper concludes that "timely screening for diabetes in women who are pregnant or planning to be pregnant provides a window of opportunity to prevent CAs in offspring.". However it is unclear from this analysis how such prevention would occur. Was there any data in the studies about how well controlled the blood glucose levels were in the mothers - comparing studies with well controlled vs less well controlled would be extremely useful to determine the possibility of prevention . As the authors make statements about the possibility of prevention they need to investigate this in their data. If it is not possible it should be flagged up as an area that needs further work.

5. I think it is not correct to include all categories of exposed women in the same analysis eg in Fig E. GDM,PGDM,T1 Diabetes and T2 Diabetes are all included and an overall estimate is produced. There is in expectation differences between PGDM and GDM so it is not relevant to combine them. 4 separate plots will make it easier to see how each individual study varies from the combined estimate for all the studies with the same exposure. I think analysing the studies separately may also slightly alter the estimates as the distribution of variances will be altered. The x-axis scale should also be more informative than just 0.1,0 and 10. Consider how to display study by "Oliveria-Brancali" as it is clearly a small study and is causing all other studies to appear very compact.

6. Forest plots are an important part of systematic reviews and I therefore suggest that revised Figure E is included in the main manuscript.

7. I would prefer to see the forest plots for the different anomalies rather than the funnel plots in figure F

Minor Comments

Table 1 : Years should be categorised Before 2000, 2000-2009 and In or after 2010

Table 2 and Table 3: would it be possible to give the numbers of each specific defect in order to have a feeling for how large the samples are 

Reviewer #4: Briefly, my comments are as follows: This is a meta-analysis on a very large population of pregnant women with PGDM and GDM, assessing the rate of major congenital malformations (congenital anomalies) and cardiac malformations among the offspring of these mothers. The findings are important, but not new. However, since the studied population is very large and the study is well designed, I think that it merits publication in the journal. I have several comments related to the description of embryologic and teratologic issues. First, the authors use "time post conception" yet they call that "time of pregnancy". The usual definition of pregnancy is from the last menstrual period while fertilization generally occurs about two weeks later. Hence, they should state that their definition of pregnancy is from the time of fertilization. In addition, when they describe the heart development (i.e. lines 394-399) they should state the weeks when the cardiac septum develops (weeks 4-7 post fertilization) not just "first 2 months" that is simply incorrect. It is expected to be more careful with embryological and teratological descriptions

The authors make no attempt to explain the increased rate of anomalies in GDM and the differences between PGDM and GDM. This is perhaps not mandatory, but mentioning several times the need for screening for diabetes in pregnant women as a means to prevent the increased rate of malformations need explanations. They should explain that this will enable better glycemic control which might reduce the rate of malformations, especially during organogenesis, and discuss several studies in this direction. The tables and figures are OK. Thanks. Asher Ornoy, MD .

[LINK]

---

## [Decision Letter · Decision Letter 2]

14 Dec 2021

Dear Dr. Gao,

Thank you very much for re-submitting your manuscript "Risks of specific congenital anomalies in offspring of women with diabetes: a systematic review and meta-analysis of population-based studies of over 80 million births" (PMEDICINE-D-21-01554R2) for review by PLOS Medicine.

I have discussed the paper with my colleagues and the academic editor and it was also seen again by two reviewers. I am pleased to say that provided the remaining editorial and production issues are dealt with we are planning to accept the paper for publication in the journal.

[LINK]

We look forward to receiving the revised manuscript by Dec 21 2021 11:59PM.   

Sincerely,

Caitlin Moyer, Ph.D.

Associate Editor 

PLOS Medicine

plosmedicine.org

Requests from Editors:

Comments from Reviewers:

1. Title: Please revise the title slightly, and please capitalize the first word of the subtitle, after the colon. We suggest: “Risks of specific congenital anomalies in offspring of women with diabetes: A systematic review and meta-analysis of population-based studies including over 80 million birth”

2. Financial disclosure: Please state whether any sponsors or funders (other than the named authors) played any role in: Study design, Data collection and analysis, Decision to publish, Preparation of the manuscript. If sponsors or funders had no role in the research, please include this sentence: “The funders had no role in study design, data collection and analysis, decision to publish, or preparation of the manuscript.”

3. Data availability statement: Please revise the data availability statement entered into the manuscript submission system, as this is the statement that will be used in the event of publication. This currently reads “All relevant data will be made available upon request to the corresponding author via e-mail.” Please change this to:“The metadata underlying the reported analyses have been deposited in Zenodo (DOI: 10.5281/zenodo.5595068).” or similar, and please check that the DOI is correct.

4. Metadata: In the online repository of excel files, it would be helpful to include a “readme” type of file in the repository, explaining the meaning of the column headers in each spreadsheet.

5. Abstract: Lines 53-57: “Of the 23 type-specific CAs of other systems in offspring, maternal PGDM was associated with a significantly increased risk of CAs in 21 categories; the corresponding RRs ranged from 1.57 (for hypospadias, 95% CI: 1.22 to 2.02; P < 0.001) to 18.18 (for holoprosencephaly, 95% CI: 4.03 to 82.06; P = 0.085).” It should be noted that it does not seem to be the case that the RR for holoprosencephaly reached statistical significance, though the sentence is worded as if it does. 

The same wording inconsistency is present at Lines 57-60. “Maternal GDM was associated with a small but significant increase in the risk of CAs in nine categories; the corresponding RRs ranged from 1.14 (for limb reduction, 95% CI: 1.06 to 1.23; P = 0.866) to 5.70 (for heterotaxia, 95% CI: 1.09 to 29.92; P = 0.008).” We suggest revising to make it clear that you are reporting 1) the number of significant associations between either PGDM or GDM and type specific CAs out of the total type-specific CAs, and then 2) reporting the highest and lowest corresponding RRs out of the total of 23 type-specific CAs.

6. Abstract: Lines 53-57: Please revise the description of numbers of type specific CAs and CHD examined, as the Author Summary and Discussion mention 42 categories assessed.

7. Abstract: Line 67; We suggest removing “with no overlap in the 95% CIs”

8. Abstract: Line 67-69: Your findings do not directly inform on the causality of the relationship. We suggest revising to temper this conclusion. It may be helpful to focus the sentence more on how screening may inform clinical care and identifying risk.

9. Author summary: Please avoid directly copying text from the abstract for the author summary (for example, at Lines 72-75). Please revise to make this section distinct. Please see our author guidelines for more information: https://journals.plos.org/plosmedicine/s/revising-your-manuscript#loc-author-summary

10. Author summary: Lines 83-85: The text here implies there are 42 categories of CAs, while the abstract describes 23 type-specific CAs. Please revise to resolve the discrepancy.

11. Author summary: Lines 84-85: Please revise to: “while maternal gestational diabetes is associated with a small but significant increase in the risk of CAs…”

12. Author summary: Lines 97-98: We suggest removing this point, as the study does not examine insulin treatment for blood glucose control during pregnancy.

13. Introduction: Line 116: As mentioned by a reviewer, please clarify if the meaning is “third to eighth week of gestation” in this sentence.

14. Methods: Line 142: Please include a copy of the registered protocol as supporting information, and please refer to it here (e.g. S1_Protocol).

15. Methods: Line 164: Please reference the PRISMA flowchart (e.g. Fig 1, PRISMA flowchart).

16. Methods: Line 175: Please revise to clarify what specifically is meant by “overall CAs and CHD” and type-specific CAs for primary and secondary outcomes (e.g. rates).

17. Results: Line 274: Please clarify the p values reported for results, for example, whether or not the risk of overall CAs in offspring of women with type 2 diabetes was statistically significant or not. Please check if the reference to Fig J1 here should be Fig L1.

18. Results: Please clarify the p values reported for tests of significance of relative risks throughout this section, as in some cases there seems to be discrepancy between the 95% CIs and p values given.

19. Results: Please check all references to the figures and tables located in S1 Fig, to ensure accuracy.

20. Discussion: Line 354: Please revise to “...while maternal GDM was associated with a small but significant increase in risk of…”.

21. Discussion: Line 395-397: As mentioned by a reviewer, please use consistent language when referring to stages of pregnancy. In this paragraph, it is not clear if “22 days” is in reference to fertilization or LMP, for example. At line 397, please clarify if this should be “4-7 weeks gestation” or similar.

22. Discussion: Line 417: Please fully define “CAKUT” in this sentence.

23. Discussion: Line 432: Please check here and throughout that the number “45” is consistently the number of type-specific CAs examined (elsewhere, the number 42 is mentioned).

24. Discussion: Line 473: We suggest tempering this statement slightly, such as: “screening for diabetes in pregnant women may enable better glycemic control, and may enable identification of women and offspring at risk for CAs.” or similar.

25. Table 1: Please indicate in the legend that “No” refers to “Number” within the table.

26. Line 542: Please remove the Author Contributions, Competing Interests, Data Availability Statement, and Funding from the main text and please ensure all information is completely and accurately entered in the relevant fields of the manuscript submission system.

27. Figure 6: Please clarify “All P meta-regression ˂ 0.001 for pre-gestational diabetes versus gestational diabetes.” to indicate that this is true for both the comparison within congenital anomalies and congenital heart defects between gestational diabetes and pre-gestational diabetes. Please indicate in the legend if there are any other comparisons to report.

28. Table E and F within S1 Table file: Please move these Tables to the main text of the manuscript.

29. Supporting Information files: The supporting information name and number are required in a caption, and we highly recommend including a one-line title as well. You may also include a legend in your caption, but it is not required. Format your supporting information captions as follows: S1 Text. Title is strongly recommended. Legend is optional.

30. S1 Text: We suggest the search strategy and each checklist be provided as its own supporting information file.

31. PRISMA Checklist: Please revise items 1 and 2 of the PRISMA checklist, referring to Title and Abstract rather than page numbers. We note that there is no assessment of certainty of evidence. We suggest reporting on certainty of each outcome described in the results, in light of the bias assessment reported at line 255.

Reviewer #2: Many thanks authors for their great effort to improve the manuscript. I am satisfied with the response and revision. No further issues needing attention.

Reviewer #3: The authors have addressed all my earlier comments

[LINK]

---

## [Editor Report · Decision Letter 3]

21 Dec 2021

Dear Dr. Gao,

Thank you very much for re-submitting your manuscript "Risks of specific congenital anomalies in offspring of women with diabetes: A systematic review and meta-analysis of population-based studies including over 80 million birth" (PMEDICINE-D-21-01554R3) for review by PLOS Medicine.

Provided the remaining editorial and production issues are dealt with we are planning to accept the paper for publication in the journal.

The remaining issues that need to be addressed are listed at the end of this email. Please take these into account before resubmitting your manuscript:

[LINK]

In revising the manuscript for further consideration here, please ensure you address the specific points made by the editors. In your rebuttal letter you should indicate your response to the editors' comments and the changes you have made in the manuscript. Please submit a clean version of the paper as the main article file. A version with changes marked must also be uploaded as a marked up manuscript file.

We expect to receive your revised manuscript within 2 weeks. Please email us (plosmedicine@plos.org) if you have any questions or concerns.

We look forward to receiving the revised manuscript by Jan 04 2022 11:59PM.   

Sincerely,

Caitlin Moyer, Ph.D.

Associate Editor 

PLOS Medicine

plosmedicine.org

Requests from Editors:

1. Data Availability Statement: Please include the complete link to the study data in the Data Availability Statement: https://doi.org/10.5281/zenodo.5783967

2. Title: Please change the title to: “Risks of specific congenital anomalies in offspring of women with diabetes: A systematic review and meta-analysis of population-based studies including over 80 million births”

3. Abstract and Results: Thank you for clarifying the reporting of the p values from the tests for heterogeneity. Please report the relative risks together with both 95% CIs and p values for results described in the text. Reporting of the results for the tests of heterogeneity in the text provides additional useful information.

4. Throughout: CA risk categories. In the abstract at line 52, the total number of type specific CA categories is indicated as 23. However, in the author summary and the discussion, 45 categories are mentioned. Please clarify in the Abstract that the 23 includes the CA categories that exclude the CHD-related categories.

5. Abstract: Lines 47-50: Please include the p values associated with the increase in risk for overall CA/CHD in women with PGDM or GDM.

6. Results: Lines 265-274: Please include the p values for RR for CAs associations with PGDM, type 1 diabetes, type 2 diabetes (p value associated with heterogeneity test may be reported separately).

7. Results: Lines 278-288: Please include the p values for RR for CHD associations with PGDM, type 1 and type 2 diabetes (p value associated with heterogeneity test may be reported separately).

8. Results: Lines 290-300: Please include p values for tests for RR for GDM associations with CHD (p value associated with heterogeneity test may be reported separately).

9. Results: Lines 303-314: Please include the p values for the RR reported for GDM and PGDM associated with type-specific CAs (p value associated with heterogeneity test may be reported separately).

10. Lines 553-555: Please remove the Financial Disclosure section from the main text, and please make sure all information is accurately entered in the Funding section of the manuscript submission system. You note here that the funders had no role in the study. However, the Funding section of the manuscript notes that “The authors received no specific funding for this work.” Please clarify.

11. Figures 2, 3, 4, and 5: Please report the overall RR with 95% CIs and p values. Please define “DL” in the legends.

12. Figure 6: Please report the p value in addition to the RR and 95% CIs.

13. Tables 2, 3, 4, and 5: Please report the p values in addition to the 95% CIs for the pooled RRs.

14. Figures G - L in S1 Fig: Please note the overall RR with 95% CIs and p values. Please define the abbreviation “DL” in the legends.

15. Line 58-60, Line 330-332, Line 435-437 and Line 442-444: There seems to be some overlap in text with https://doi.org/10.1136/bmj.m3222. Please revise to avoid this.

[LINK]

---

## [Editor Report · Decision Letter 4]

22 Dec 2021

Dear Dr Gao, 

On behalf of my colleagues and the Academic Editor, Jenny Myers, I am pleased to inform you that we have agreed to publish your manuscript "Risks of specific congenital anomalies in offspring of women with diabetes: A systematic review and meta-analysis of population-based studies including over 80 million births" (PMEDICINE-D-21-01554R4) in PLOS Medicine.

Also, please address the following editorial request:

1. Methods: Line 224: Please add the criteria by which statistical significance was evaluated for the pooled RR reported (e.g. "Evidence for statistical significance for each pooled RR was based on whether or not the 95% confidence intervals for the RR included the null value of 1." or similar).

PRESS

Sincerely, 

Caitlin Moyer, Ph.D. 

Associate Editor 

PLOS Medicine